# Realtime measurement of phase partitioning of organic compounds using a Proton-Transfer-Reaction Time-of-Flight Mass Spectrometer coupled to a CHARON inlet

**Yarong Peng[1,2], Hongli Wang[2,*], Yaqin Gao[2], Shengao Jing[2], Shuhui Zhu[2], Dandan Huang[2], Peizhi Hao[3], Shengrong Lou[2], Tiantao Cheng[4,5,*], Cheng Huang[2], Xuan Zhang[3,*]**

[1] Department of Environmental Science and Engineering, Fudan University, Shanghai, 200438, China

[2] State Environmental Protection Key Laboratory of Formation and Prevention of Urban Air Pollution Complex, Shanghai Academy of Environmental Sciences, Shanghai, 200233, China

[3] School of Natural Sciences, University of California, Merced, 95343, USA

[4] Department of Atmospheric and Oceanic Sciences, Fudan University, Shanghai, 200438, China

[5] Big Data Institute for Carbon Emission and Environmental Pollution, Fudan University, Shanghai, 200438, China

*Correspondence to:* Hongli Wang (wanghl@saes.sh.cn), Tiantao Cheng (ttcheng@fudan.edu.cn), Xuan Zhang (xzhang87@ucmerced.edu)

**Abstract.** Understanding the gas-particle partitioning of semivolatile organic compounds (SVOCs) is of crucial importance in the accurate representation of the global budget of atmospheric organic aerosols. In this study, we quantified the gas- vs. particle-phase fractions of a large number of SVOCs in real time in an urban area of East China with the use of a CHemical Analysis of aeRosols ONline (CHARON) inlet coupled to a high resolution Proton Transfer Reaction Time-of-Flight Mass Spectrometer (PTR-ToF-MS). We demonstrated the use of the CHARON inlet for highly efficient collection of particulate SVOCs while maintaining the intact molecular structures of these compounds. The collected month-long dataset with hourly resolution allows us to examine the gas-particle partitioning of a variety of SVOCs under ambient conditions. By comparing the measurements with model predictions using the instantaneous equilibrium partitioning theory, we found that the dissociation of large parent molecules during the PTR ionization process likely introduces large uncertainties to the measured gas- vs. particle-phase fractions of less oxidized SVOCs, and therefore, caution should be taken when linking the molecular composition to the particle volatility when interpreting the PTR-ToF-MS data. Our analysis suggests that understanding the fragmentation mechanism of SVOCs and accounting for the neutral losses of small moieties during the molecular feature extraction from the raw PTR mass spectra could reduce, to a large extent, the uncertainties associated with the gas-particle partitioning measurement of SVOCs in the ambient atmosphere.

## 1. Introduction

Gas-particle partitioning of semivolatile organic compounds (SVOCs) is a critical process involved in the formation and evolution of atmospheric organic aerosols (OA). Traditionally,

gas-particle equilibrium partitioning of organic substances is assumed to be established
instantaneously (Zhang and Seinfeld, 2013), this assumption is in question if particles are semi-
solid or glassy (Shiraiwa et al., 2013). Most studies to date addressing the kinetic limitations in
partitioning have used indirect and/or theoretical methods that are lack of chemical and
molecular specificity (Mai et al., 2015; Shiraiwa and Seinfeld, 2012). Direct measurements of
gas-particle partitioning of SVOCs are needed in order to develop accurate parameterizations
for the organic aerosol formation in climate models.

The major challenge in the characterization of gas-particle partitioning of SVOCs lies in

the realtime measurement of labile compounds while maintaining their intact molecular
structures with minimal fragmentation (Zhang et al., 2016a; Zhang et al., 2016b; Zhang et al.,
2019). In recent years, soft ionization techniques coupled to mass spectrometry have been
widely used for the measurement of gas-phase SVOCs at the molecular level (Veres et al., 2008;
Crounse et al., 2006; Heald and Kroll, 2020). Combined with thermal desorption methods, these
techniques have also been deployed to measure organic compounds in both gas and particle
phases nearly simultaneously (Krechmer et al., 2016). A notable example would be the use of
the Filter Inlet for Gases and AEROsols coupled with the Chemical Ionization Mass
Spectrometry (FIGAERO-CIMS) to quantify the gas-particle partitioning of a broad range of
organic compounds in real time (Lopez-Hilfiker et al., 2014; Ye et al., 2021; Voliotis et al., 2021;
Wang et al., 2020a; Lutz et al., 2019; Lee et al., 2018; Le Breton et al., 2018; Stark et al., 2017;
Lopez-Hilfiker et al., 2016; Lopez-Hilfiker et al., 2015; Palm et al., 2020). A number of studies
among those have reached a consensus that the thermograms method, i.e., using the calibrated
thermal desorption profiles vs. temperature to derive the volatility, likely provides the best
estimates of the actual phase distribution. In contrast, using the directly measured gas- and
particle-phase fractions of a given analyte will most likely introduce a significant positive or
negative bias to the volatility estimation due to the thermal decomposition of labile organic
compounds during the desorption process (Lopez-Hilfiker et al., 2015; Stark et al., 2017). Such
thermal decomposition (or ion fragmentation) artifacts, either positive or negative depending
on the molecular size, have been suggested to constitute the largest uncertainties in the
estimation of phase partitioning behaviors of SVOCs using the thermal desorption method at
ambient pressure (Thompson et al., 2016).

Along with the line of thermal desorption method development, an inlet designed for the

CHemical Analysis of aeRosols ONline (CHARON) has been developed and coupled to the
Proton Transfer Reaction Time-of-Flight Mass Spectrometer (PTR-ToF-MS) in recent years.
As CHARON-PTR-ToF-MS does not rely on any form of pre-concentration on surfaces, it
could provide online and direct measurements of organic compounds in both phases, compared
with traditional thermal desorption instruments which still need to address artifacts during the
particle collection and desorption processes. Another potential advantage of CHARON-PTR-
ToF-MS is that the chemical information of the collected particles can be studied qualitatively
and quantitatively over a chemical composition level even at sub-nanogram mass
concentrations per molecule owing to the well studied ion-molecule reaction chemistry in PTR-
ToF-MS (Piel et al., 2019). CHARON has shown promising potential in the realtime analysis
of the chemical composition and spatiotemporal distributions of aerosols with laboratory-,
ground-, and aircraft-based platforms (Piel et al., 2019; Tan et al., 2018; Gkatzelis et al., 2018a;
Gkatzelis et al., 2018b; Muller et al., 2017; Eichler et al., 2017; Eichler et al., 2015; Antonsen
et al., 2017; Leglise et al., 2019; Piel et al., 2021). As a relatively new technique, the use of
CHARON-PTR-ToF-MS to investigate the gas-particle partitioning of organic compounds is
still quite limited. Only one study by Gkatzelis et al. (2018b) deployed CHARON, together
with two other aerosol sampling inlets, to measure the OA formation and aging from
monoterpenes and real plant emissions in chamber experiments. Whether the CHARON inlet
can be applied to the study of gas-particle partitioning of organic compounds under the actual
atmospheric conditions remains to be validated.
In this study, we assess the applicability of the CHARON inlet to the time-resolved
collection of organic compounds in their native molecular state using laboratory tests with a
series of authentic standards. We further employ the CHARON inlet coupled to a high
resolution PTR-ToF-MS instrument to measure an array of gaseous and particulate SVOCs in
an urban area of East China. The obtained month-long hourly dataset allows us to examine the
gas-particle partitioning of SVOCs spanning a range of volatilities. By comparing the
measurements with model predictions using the instantaneous equilibrium partitioning theory,
we found that fragmentation during the PTR ionization process may introduce large
uncertainties to the measured gas- vs. particle-phase fractions of less oxidized SVOCs.
Understanding the dissociation patterns of parent molecules and accounting for the
fragmentation losses when extracting the molecular features from the raw PTR mass spectra
are needed to improve the measured accuracy of SVOCs partitioning between the gas and
particle phase.

**2.  Material and Methods**
**2.1. Sampling site**
The sampling site located in the campus of the Shanghai Academy of Environmental
Science (SAES) is representative of a typical urban setting surrounded by restaurants, shopping
malls, and residential and commercial buildings (Fig. S1). Two traffic-heavy streets in the area
(Caobao Road and Humin Highway) are located ~ 150 m and 450 m lateral distances to the east
of the sampling site. A few number of petrochemical and chemical industrial facilities are
located ~ 50 km to the south and southwest of the observation site, which likely bring certain
long-lived pollutants to the site at a typical wind speed of ~ 1 – 3 m/s. Major pollution sources
in the area include traffic, commercial and residential activities, and regional transport (Huang
et al., 2021; Peng et al., 2023). The sampling inlet of the PTR-ToF-MS instrument was installed
on the roof of an eight-story building ~ 24 m above the ground. A comprehensive measurement
of gas- and particle-phase compounds in the ambient air was performed from Oct 24 to Nov 22.
During the sampling period, the average temperature, relative humidity, and wind speed were
$18.0 \pm 3.0$ °C, $61.0 \pm 15.0\%$, and $1.8 \pm 0.7$ m/s, respectively. The prevailing wind direction in
this region was from the northwest and north during the polluted period (Fig. S2).
**2.2. CHARON-PTR-ToF-MS**
**2.2.1.   Operation protocols**

A Proton Transfer Reaction Time-of-Flight Mass Spectrometer (PTR-ToF-MS) coupled to

a CHemical Analysis of aeRosols ONline (CHARON, Ionicon Analytik Inc, Innsbrunck,
Austria) inlet was employed to measure the gas- and particle-phase concentrations of a series
of SVOCs. The PTR-ToF-MS instrument used here is equipped with a radio frequency (RF)-
only quadrupole ion guide that transmits ions more efficiently (PTR-QiTOF, Ionicon Analytik
Inc) but results in a low-mass cutoff (Fig. S4). The operating parameters of the PTR-ToF-MS
were held constant during the entire measurement period. The drift tube pressure, temperature,
and voltage were 2.9 mbar, 120 °C, and 500 V, respectively. These conditions correspond to an
$E/N$ ($E$ is the electric field, and $N$ is the number density of the gas molecules in the drift tube)
value of ~ 100 Td (1 Td = $10^{-17}$ V $cm^2$) and a reaction time of 120 μs. Note that the $E/N$ value
determines the collision energy of ions in the reactor and therefore the degree of fragmentation
and cluster formation. The operating conditions were selected for the purpose of relatively low
fragmentation intensities (compared to 120 – 140 Td) and limited production of water clusters
(compared to 60 – 80 Td). Low $E/N$ enhances the degree of water clustering, which complicates
the analysis of analyte ions due to a complex interplay between cluster formation ($RH^+(H_2O)_n$)
and proton transfer reactions (Holzinger et al., 2019). During the campaign, the sensitivity of
the PTR-ToF-MS was in the range of 300 – 1000 ncps $ppb^{-1}$ and the mass resolution was
maintained at ~ 5000 m/Δm. Mass spectra were collected at a time resolution of 10 s.
The CHARON inlet consists of 1) a gas-phase denuder (GPD) for stripping off gas-phase
analytes, 2) an aerodynamic lens (ADL) for particle collimation which is combined with an
inertial sampler for emanating the particle-enriched flow, and 3) a thermo-desorption unit (TDU)
for particle volatilization. The CHARON inlet functionality has been described in great detail
by Eichler et al. (2015). The inlet we used here had a particle enrichment factor of ~ 15, as
discussed shortly. The vaporizer (TDU) was operated at 140 °C and ~ 8 mbar absolute pressure.
This temperature was chosen to ensure all the unknowns observed in the field can be evaporated
effectively while maintaining relatively intact molecular structures, see more details in Section
3.1. Measurements of organic compounds in the gas and particle phase were conducted using a
parallel sampling system with two independent pumps, allowing for the selection of flow rates
specifically adjusted for each phase, resulting in the overall residence time of less than 2 s (Fig.
S3).
**2.2.2.    Sampling alternation between gas and particle phase**
Gas-phase compounds were measured by directly sampling the ambient air via a 2 m long
perfluoro-alkoxy (PFA) tube (1/4'' OD) capped with a polytetrafluoroethylene (PTFE) filter
(Mitex$^{TM}$ PTFE membrane, 5 μm pore size, 47 mm diameter) to prevent the clogging of
particles in the PTR capillaries. The gas-phase inlet was independently connected to the PTR-
ToF-MS instrument upstream of the drift tube via a pressure-controlled subsampling PEEK
capillary (1/16'' OD). Zero measurements were performed by overflowing catalytically
(platinum at 370°C) purified air through the inlet. Ambient particles were sampled through a
stainless steel tube (3/8'' OD) with a flow rate of ~ 3 L min$^{-1}$, out of which a flow of ~ 500 ml
min$^{-1}$ was directed to the CHARON inlet. A PM$_{2.5}$ cyclone was installed in front of the sampling
line to remove coarse particles (> 2.5 μm). The particle-phase background was measured by
placing a High-Efficiency Particulate Air filter (HEPA, HEPA-CAP 7, GE Healthcare UK
Limited, Buckinghamshire, UK) upstream of the CHARON inlet. Servo motor activated valves
made of passivated stainless steel were used for switching between the two inlet configurations.
During the campaign, CHARON-PTR-ToF-MS automatically switched between gas and
particle phase every 15 min. Detailed setup is given in Fig. S3 in the Supporting Information.
The built-in PTR-manager software (Ionicon Analytik GmbH, Innsbruck, Austria) offers
the possibility to program sequences by which the instrument switches between different
settings. It takes ~ 1 min for gases and particles to re-equilibrate when switching between these
two modes. Data generated during this transition period (~ 2 min) were not considered.
Instrument background was measured for 15 min every 5h. The limits of detection (LoD) at 1
min resolution were in the range of 5.6 ± 2.9 ng m$^{-3}$ for gases and 0.7 ± 0.5 ng m$^{-3}$ for particles,
respectively (Fig. S5). Concentrations of gaseous and particulate compounds shown here
included the last 5 min of every gas/particle-phase working mode, in order to minimize the
interferences carried over from the previous working mode by allowing for a sufficient amount
of equilibration time in the inlet (Piel et al., 2021). In order to synchronize the gas- and particle-
phase data to calculate gas-particle partitioning, the average hourly data were then used for
further analysis.

### 2.2.3. Sensitivity and calibration

Weekly calibrations were performed using a multicomponent calibration gas standard
(Linde, USA) at five concentration levels from 0.5 to 10 ppb (Fig. S4a). The calibration mixture
includes methanol, acetonitrile, acetaldehyde, acrolein, acetone, isoprene, methyl vinyl ketone,
methyl ethyl ketone, 2-pentanone, toluene, styrene, p-xylene, 1,3,5-trimethylbenzene,
naphthalene and α-pinene. Calibration standards with higher molecular weight were excluded
because we only considered ion masses below 200 amu from the field measurement for the
study of gas-particle partitioning, see discussions given in Text S1. Here the sensitivity of PTR-
ToF-MS is defined as the normalized ion intensity of $RH^+$ (ncps) obtained at a mixing ratio of
1 ppb. For a given species $(R)$, its sensitivity $(S$, ncps ppb$^{-1})$ is a linear function of the rate
constant of its reaction with $H_3O^+$ $(k)$ :

$$S = \frac{\frac{I_{RH^+}}{I_{H_3O^+}} \times 10^6}{\frac{[R]}{N} \times 10^9} = k \times N \times 10^{-3} \times t \times \frac{T_{RH^+}}{T_{H_3O^+}} \times F_{RH^+} \tag{1}$$

$$\text{corrected } S = \frac{S}{\frac{T_{RH^+}}{T_{H_3O^+}} \times F_{RH^+}} = a \times k \tag{2}$$

where the signals of $H_3O^+$ $(I_{H_3O^+})$ and $RH^+$ ions $(I_{RH^+})$ measured by the mass analyzer (in cps)
can be related to the signals of $H_3O^+$ $([H_3O^+])$ and $RH^+$ $([RH^+])$ ions at the end of the drift tube,
using their respective transmission efficiencies $(T_{H_3O^+}$ and $T_{RH^+})$ from the drift tube to the
detector (De Gouw and Warneke, 2007). $[R]$ is the concentration of species $R$ and $N$ is the
number density of gas in the drift tube. The reaction time $(t)$ is determined by the ion drift
velocity. $F_{RH^+}$ represents the fraction of product ions detected as $RH^+$ ions ($0 \leq F_{RH^+} \leq 1$). For
non-fragmenting compounds, $F_{RH^+} = 1$. The measured sensitivity is further corrected by
accounting for fragmentation and transmission efficiency, the value of which was derived from
laboratory experiments (Fig. S4a). Uncertainties associated with the addition of the low-mass
filter have been accounted for in the regression of individual transmission efficiency
measurements on corresponding mass to charge ratios. The overall relative standard deviations
were less than 15%. *a* is the slope of the linear regression of the corrected sensitivities on the
proton-transfer-reaction rate coefficients (*k*), as shown in Fig. S4b. Following the method by
Sekimoto et al. (2017), the linear regression result was used to determine the sensitivities of all
uncalibrated species. The overall uncertainty was less than 15% for compounds with standards
and around 50% for those without standards. Calculated sensitivity based on this method agrees
well with measurements of authentic standards (Fig. S4c).
**2.2.4. Enrichment factor**
The CHARON inlet was calibrated routinely with pure ammonium nitrate particles to
derive the enrichment factor as a function of the particle size following the procedures described
in Eichler et al. (2015). In addition, we used a selection of authentic standards (Table S1) to test
the effect of desorption temperature on the enrichment factor of labile compounds. Previous
studies with CHARON generally used a temperature of 140 °C to vaporize particles (Leglise et
al., 2019; Gkatzelis et al., 2018b; Tan et al., 2018). Herein, we tested the TDU temperature
ranging from 70 to 140°C. The selected chemical standards were individually dissolved in
distilled water (ethanol in the case of 2-Pentadecanone and 1-Pentadecanol) and nebulized by
an atomizer (TSI 3076, Shore-view, MN, USA) that was pressurized with ultrapure zero air.
The nebulizer outflow was diverted through two diffusion dryers to remove water vapor and an
activated charcoal denuder (NovaCarb F, Mast Carbon International Ltd., Guilford, UK) to
remove organic vapors. The resulting flow of polydisperse particles was then delivered into a
differential mobility analyzer (DMA, TSI 3080) for particle size selection. The transmitted
particles at a given size bin (300 nm for organics and $100 - 450$ nm range for ammonium nitrate)
were introduced into the CHARON-PTR-ToF-MS analyzer and a condensation particle counter
(CPC, TSI 3775), respectively. Particle mass concentrations were calculated based on the CPC
number distribution measurements by assuming a shape factor of 0.8 for ammonium nitrate
particles and 1 for organic particles, respectively.

The particle enrichment factor (*EF*) of a given analyte $i$ was calculated as the ratio of the

PTR-ToF-MS derived vs. CPC derived mass concentrations of analyte $i$ at a given particle size
bin:

$$VMR_{(PTR)i} = \frac{I_i}{S_i} \qquad (3)$$

$$VMR_{(CPC)i} = \rho_i \times V \times N_i \times V_m / Mw_i \qquad (4)$$

$$EF = \frac{VMR_{(PTR)i}}{VMR_{(CPC)i}} \qquad (5)$$

where $I_i$ is the normalized signal of species $i$ (ncps) by PTR-ToF-MS, $S_i$ is the sensitivity
(ncps ppb$^{-1}$), *VMR* is the volume mixing ratio (ppb), $\rho$ is the density of species $i$ (g cm$^{-3}$),
$V$ is the volume of a particle sphere (m$^3$), $N_i$ is the number concentration of particles measured
by CPC (cm$^{-3}$), $V_m$ is the molar volume of an ideal gas at 1 atm (22.4 L mol$^{-1}$), $Mw_i$ is the
molecular weight (g mol$^{-1}$). As the calculated sensitivities of most organics in the absence of
authentic standards are subject to uncertainties (15% – 50%), we will herein use the
multiplication of *EF* and $S_i$ to evaluate the combined effect of CHARON enrichment and
sensitivity on the measured concentrations of a given analyte $i$ in the particle phase.

### 2.2.5. Data processing

Data were analyzed using the Tofware package (v3.2.0, Tofwerk Inc), within the Igor Pro software (v7.0, Wavemetrics). Using this package, time-dependent mass calibrations were performed using four ions ($H_3^{18}O^+$, $NO^+$, $C_6H_5I^+$ and $C_6H_5I_2^+$), where $C_6H_5I^+$ and $C_6H_5I_2^+$ were produced from the internal standard di-iodobenzene. The relative mass deviation was within 6 − 8 ppm across the mass spectra. Considering the humidity dependence of reagent ions ($H_3O^+$ and $H_3O^+(H_2O)$), the fitted product ion signals ($RH^+$) were normalized to a standard reagent ion of $10^6$ cps (counts per second). Elemental composition was determined based on the accurate *m/z* (mass to charge ratio) and isotopic pattern analysis. A list of ~ 1600 ions was extracted, including both gas- and particle-phase ions. Molecular formulas including only C, H, and O atoms were assigned to the detected ions by the addition of one proton in cases where the elemental composition analysis returned multiple options. About 85% of the signals were elementally resolved by the $C_xH_yO_z$ formula in ambient air mass spectra. A small number of nitrogen containing compounds, such as nitroaromatics, were also identified but not included in the following analysis. Throughout of the context, we use the word "species" to refer to all compounds with assigned molecular formula, which may include multiple isomers.

### 2.3. Complementary measurements

In addition to CHARON-PTR-ToF-MS, a Thermal desorption Aerosol Gas chromatograph (TAG) was also employed to measure a series of particle-phase organic species. Details of the TAG operation and data analysis protocols can be found in previous studies (He et al., 2020; Wang et al., 2020b; Zhu et al., 2021). The elemental composition and mass concentration of particles were measured by an Aerodyne high-resolution time-of-flight Aerosol Mass Spectrometer (AMS), with details of operation and quality control protocols given by our recent

study (Huang et al., 2021). Volatile organic compounds (VOCs, $C_2 - C_{12}$) were analyzed by a custom-built online gas chromatography system equipped with a mass spectrometer and a flame ionization detector (GC-MS/FID). The performance of this system can be found in our previous publications (Zhu et al., 2018; Wang et al., 2014). Meteorological parameters (ambient temperature, wind speed, wind direction, and relative humidity) were collected by an automatic weather station (Metone 590 series) mounted on the roof top of the campaign site.

**2.4. Gas-particle partitioning measurements vs. modeling**

The CHARON-PTR-ToF-MS measured gas- and particle-phase concentrations of a given species $i$ can be used to calculate its particle-phase fraction ($F_{p,i}$).

$$P_i = \frac{\frac{I_{p,i} \times \left(\frac{m}{z_i} - 1\right)}{V_m \times S_i}}{EF} \tag{6}$$

$$G_i = \frac{I_{g,i} \times \left(\frac{m}{z_i} - 1\right)}{V_m \times S_i} \tag{7}$$

$$F_{p,i} = \frac{P_i}{P_i + G_i} \tag{8}$$

where $P_i$ and $G_i$ are the mass concentrations (ng m$^{-3}$) of species $i$ in the particle and gas phase, respectively. $I_{p,i}$ and $I_{g,i}$ are the normalized signal (ncps) of the PTR-ToF-MS detected ion $i$ in the particle and gas phase, respectively. $V_m$ is taken as 22.4 L/mol. $S_i$ is calculated or measured sensitivity (ncps ppb$^{-1}$), see details in Section 2.2.3. As structural isomers cannot be resolved in the mass spectra, the calculation here assumes that all isomers with the same molecular formula have the same chemical properties, i.e., saturation vapor pressures. Substitution of Equations (6) and (7) to Equation (8) yields the final expression of the particle-phase fraction of species $i$ ($F_{p,i}$), which is a function of the observed PTR-MS raw signals of species $i$ in the gas and particle phase (in total ion counts), as well as the particle enrichment

factor ($EF$) of species $i$.

$$F_{p,i} = \frac{I_{p,i}/EF}{I_{p,i}/EF + I_{g,i}} \tag{9}$$

Gas-particle partitioning of a given analyte $i$ was also modeled using the equilibrium
partitioning theory (Pankow, 1994):

$$F_{p,i} = \frac{1}{1 + C_i^* / C_{OA}} \tag{10}$$

$$C_i^* = \frac{10^6 Mw_i \zeta_i p_i}{RT} \tag{11}$$

where $C_{OA}$ is the organic aerosol concentration measured by AMS (μg m$^{-3}$), $C_i^*$ is the
saturation mass concentration (μg m$^{-3}$), $Mw_i$ is the molecular weight (g mol$^{-1}$), $\zeta$ is the
activity coefficient (assumed as unity), $p_i$ is the pure component liquid vapor pressure (Pa), $R$
is the universal gas constant ($8.2 \times 10^{-5}$ m$^3$ atm K$^{-1}$ mol$^{-1}$), and $T$ is the ambient temperature
(K). As detailed chemical information is lacking for all species detected by PTR-ToF-MS, here
we use the expression given by Donahue et al. (2011) to approximate the value of $C_i^*$:

$$\log_{10} C_i^* = (n_C^0 - n_C^i)b_C - n_O^i b_O - 2\frac{n_C^i n_O^i}{n_C^i + n_O^i} b_{CO} \tag{12}$$

where $n_C^0 = 25$, $b_C = 0.475$, $b_O = 2.3$, and $b_{CO} = -0.3$.

**2.5. Uncertainties in the measured and modeled gas-particle partitioning**
The uncertainty associated with the PTR-MS measured concentrations in both gas and
particle phases is less than 15% for compounds with chemical standards based on the optimally
fitted transmission efficiency curve. For those in the absence of standards, their PTR
sensitivities were calculated theoretically using Equations (1-2), and the uncertainty in the
calculation mainly arises from the estimation of polarizability and dipole moment of the target
molecule, which has been estimated to be within ~ 50% when only the elemental composition
of that molecule is given (Sekimoto et al., 2017). It is important to note, however, that the
uncertainty associated with the estimated PTR sensitivity has zero influence on the measured
particle-phase fraction of any given compound because the sensitivity term is essentially
canceled in the divisor function in Equation (9). The uncertainty associated with the particle
fraction of a given species $i$ derived from the PTR-MS measurements arises predominantly
from the "$EF$" term. Since the uncertainty of the measured $EF$ depends on the uncertainty of
$I_{p,i}$, we thus express the overall uncertainties of the measured gas-particle partitioning as:
$$Unc(F_{p,i}) = \sqrt{Unc(\text{EF})^2 + Unc(I_{g,i})^2} \tag{13}$$
with the calibration standards used in this study, the enrichment factor is calculated to be within
25% error ($Unc(\text{EF})$) (see detailed calculations listed in Table S2 and S3), including the effect
of wall loss inside the inlet tubing and the precision of the measurement. Huang et al. (2019)
has tested the uncertainty of wall loss ($Unc(I_{g,i})$) in this PTR-MS instrument as 28%. Therefore,
the overall uncertainty in the measured particle-phase fraction ($Unc(F_{p,i})$) was estimated as

38%.

The uncertainty associated with the modeled gas-particle partitioning arises primarily from

the uncertainty in the estimation of the saturation mass concentration ($C_i^*$) based on the method
developed by Donahue et al. (2011). In this method, the saturation mass concentration of
species $i$ is a non-linear function of the numbers of carbon and oxygen atoms in that particular
species, see Equation (12). For each generic molecular formula, i.e., $C_xH_y$, $C_xH_yO$, $C_xH_yO_2$, and
$C_xH_yO_4$, Donahue et al. (2011) have used a total of 25, 48, 18, and 10 chemical standards with
known volatilities to validate the estimated saturation concentrations, and the estimated errors
were taken as 34%, 16%, 25%, and 54%, respectively (see Table S6). As the $C_xH_yO_3$ group was
not tested, we tentatively assumed the associated errors as the same as the $C_xH_yO_4$ group. The
extent to which these uncertainties may affect the difference between measurements and model
results was discussed in detail in the Supporting Information (Fig. S10).

**3.  Results and Discussion**
**3.1. Particle enrichment: effect of desorption temperature**
Thermal desorption as a common procedure used in the chemical characterization of
organic aerosols is often susceptible to fragmentation of non-refractory compounds. Due to the
high temperature used to evaporate particles collected, labile and large molecules are inevitably
subject to fragmentation, thereby introducing large uncertainties to the measured mass and
composition of the particulate organic compounds (Lopez-Hilfiker et al., 2015; Yatavelli et al.,
2012; Zhao et al., 2013). Thermal decomposition of oxidized organic compounds has been
observed at vaporizer temperature as low as 200 °C, the lowest temperature required to vaporize
OA as reported (Stark et al., 2017). While decreasing the vaporizer temperature is necessary to
maintain the intact structure of labile molecules, low temperature (e.g., 85 °C), however, might
fail to completely evaporate the collected particles into vapors, resulting in an underestimation
of the collected OA mass (Inomata et al., 2014). Here we performed a series of sensitivity tests
to identify the optimal vaporizer temperature in the CHARON inlet for the measurements of
organic compounds in the particle phase.
Prior to the temperature sensitivity test, we have validated that the particle enrichment
factor, also known as collection efficiency and defined as the ratio of the particle mass
concentration upstream to downstream of the aerodynamic lens, does not depend on the particle
size. As shown in Fig. 1a, the measured *EF* value for ammonium nitrate particles, detected as
$NO_2^+$ produced from the nitric acid vapor, remains constant as ~ 15 in the 150 – 450 nm particle
size range. The lower values in the 100 – 150 nm size range can be explained by the lower
particle transmission efficiency in the gas phase denuder, e.g., 75% – 80% for 100 nm particles
(Eichler et al., 2015). Also, particles below 150 nm are less efficiently concentrated in the
subsampling flow after the aerodynamic lens. Therefore, we used the monodisperse particles
generated from selected organic standards at 300 nm for the temperature sensitivity test.

A number of chemical standards that are representative of alcohols, carbonyls, and

carboxylic acids and with the vapor pressure ranging from $10^{-14}$ to $10^{-1}$ Pa at 25 °C (taken from
EPA EPI Suite (2012), see values given in Table S1) were used to generate organic aerosols,
which, upon size selection at 300 nm, were directed to the CHARON inlet. Particle evaporation
occurs downstream of the aerodynamic lens in the gas phase and on the tube and orifice surface
to which submicron particles rapidly diffuse at ~ 8 mbar operating pressure. The thermal
desorption unit was designed to ensure that ammonium sulfate particles ($10^{-20}$ Pa) can be
completely evaporated (Piel et al., 2019; Eichler et al., 2015). As the desorption temperature
was varied from 70 °C to 140 °C, the intensities of all detected ions (including both parent and
fragment ions) for each organic standard analyzed were stable within 15%, as shown in Fig. 1b.
Also note that we did not observe any ions produced from decarboxylation and/or dehydration
during the particle evaporation process. This is because the relative low operation temperature
and the short heat exposure time could effectively limit any thermal dissociation of organic
molecules. This demonstrates that the parent molecule fragmentation, if any, does not occur
under the range of desorption temperature used in the CHARON inlet, but rather results from
the ionic dissociation process in the PTR ionization chamber, see more discussions in Section
3.3. We therefore used the sum total of intensities of all major ions detected as the PTR-ToF-
MS response to a given organic standard analyzed. Fig. 1c shows that the derived enrichment
factors stay constant for all compounds investigated ($Mw \sim 160 - 230$ g/mol). The relative
signals of all fragment ions were stable over the range of the desorption temperature as shown
in Fig. 2. This suggests that the desorption temperature used here, even as low as 70 °C, is
sufficient to evaporate SVOCs (volatility > $10^{-14}$ Pa at 25 °C), due to the low operating pressure
(~ 8 mbar) that significantly enhances the partitioning shift to the gas phase. One low volatility
compound, sucrose ($Mw$ is 342 g/mol and vapor pressure is $4.69 \times 10^{-14}$ Pa), has a slightly lower
enhancement factor compared with all the other organic standards tested. This is mainly due to
the intensive dehydration of the parent compound in the ionization chamber, and as a result,
only a few fragment ions were captured, resulting in a lower PTR response and thereby lower
$EF$ value calculated from Equations (3-5). The $EF$ values of sucrose also have much higher
standard deviations at all temperatures due to fragmentation (Table S3).

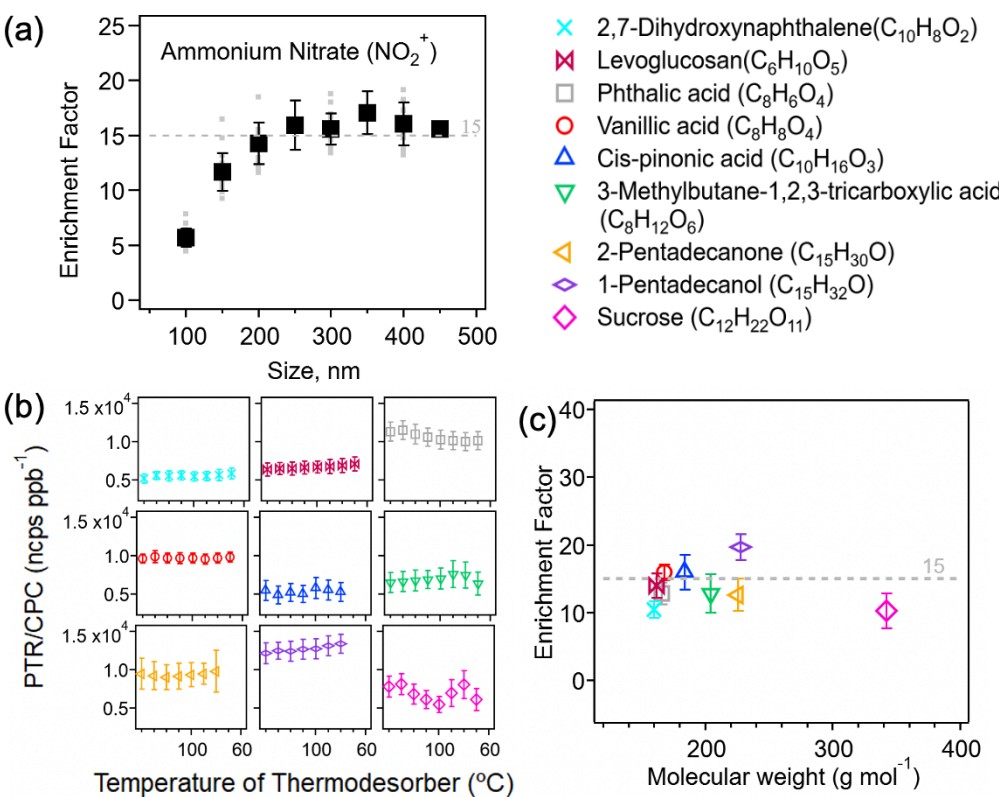


Figure 1. (a) Measured unitless enrichment factor (*EF*) of ammonium nitrate particles as a function of particle size in the 100 – 450 nm range. Grey markers represent all replicating measurements. The error bar denotes one standard deviation (1σ) of the average. (b) Ratios of PTR-ToF-MS signals (including both parent and fragment ions) to CPC counts (±1σ) at 300 nm for all organic standards studied. (c) *EF* (±1σ) of selected organic standards based on the calculated sensitivity.

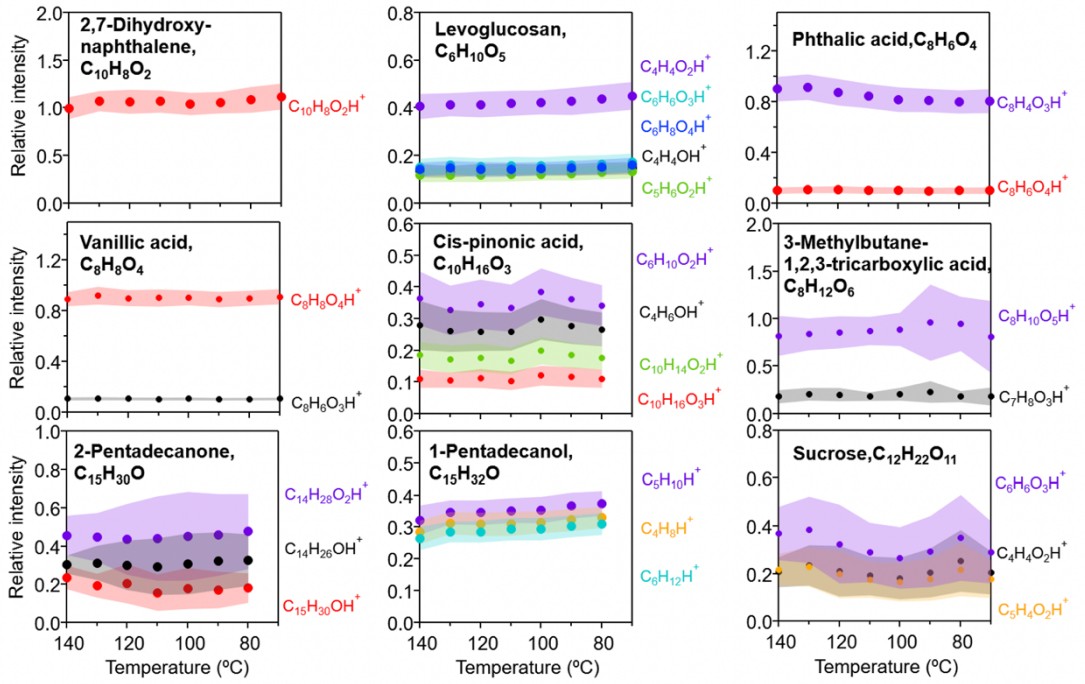

Figure 2. Ratios of CHARON-PTR-ToF-MS signals (ncps) to CPC measurements (ppb) of all detected ions (including both parents and fragments) for a given pure organic standard tested under different desorption temperatures (70 – 140 °C), normalized to the corresponding ratios obtained at 140 °C. Ions with relative intensities less than 10% are excluded. Red markers represent the parent peaks. Colored shades represent the relative standard deviations at different temperatures (exact values are given in Table S4).

It has been recognized that species with one functional group follow certain fragmentation patterns during the PTR ionization process (Pagonis et al., 2019; Francis et al., 2007; Spanel et al., 1997; Tani et al., 2003; Spanel and Smith, 1997), such as dehydration of acids and alcohols. The observed dissociation of carboxylic acid standards used in this study, e.g., phthalic acid and 3-methylbutane-1,2,3-tricarboxylic acid, can be explained by this common fragmentation

pattern. The fragmentation mechanism of muti-functionalized species is rather complicated and
a number of fragments can be produced upon PTR ionization. Nevertheless, the identity and
abundance of fragments from a given muti-functionalized species have been found comparable
under the same PTR operation protocols (Leglise et al., 2019; Gkatzelis et al., 2018a). For
example, *cis*-pinonic acid yields the following fragments (main ions only and relative
abundance in parentheses): *m/z* 71.049 (~28%), 115.075 (~36%), 167.108 (~19%), and 185.117
(~11%), which is comparable with an earlier study (Leglise et al., 2019): *m/z* 71.049 (~27%),
115.075 (~33%), 167.108 (~26%), and 185.117 (~14%) at 100 Td settings.

**3.2. Molecular features of detected organic species**
A month-long field dataset of particle- and gas-phase organic species was collected at
hourly resolution using the CHARON-PTR-ToF-MS instrument. A comparison of PTR-ToF-
MS measurements with other techniques available on site was performed for both gas and
particle phases. For the gas phase, quantitative measurements of a suite of VOCs by GC-
MS/FID, including benzene, toluene, styrene, $C_8$ and $C_9$ aromatics, acrolein, and $C_4$, $C_5$, and $C_6$
ketones, agree well with corresponding PTR-ToF-MS measurements, as shown in Fig. S6. For
the particle phase, the time series of a group of $C_xH_yO_4$ species (including $C_4H_6O_4$, $C_5H_8O_4$,
$C_6H_{10}O_4$, and $C_8H_6O_4$) are in reasonable agreement with corresponding measurements taken by
TAG (r ~ 0.60 – 0.80), although the CHARON-PTR-ToF-MS measured total molecular mass
is generally lower than the TAG measurements by a factor of 2 to 6 (Fig. S7). This is likely
caused by the fragmentation (e.g., loss of $H_2O$, see Fig. S8) of the parent compounds during the
ionization process, as discussed in detail in Section 3.3. The time series of total OA mass
characterized by CHARON-PTR-ToF-MS also agree with the AMS measurements (r ~ 0.91,
Fig. S9). Previous studies have reported the particulate organics measured by PTR-MS with a
thermal desorption inlect account for 25% – 60% mass of the total organic aerosols measured
by AMS (Holzinger et al., 2013). Direct comparison of the total OA mass loading is not
applicable here since the CHARON-PTR-ToF-MS measurement only focused on compounds
with the mass to charge ratio below 300 Th. That the majority of ions detected by PTR are
present in the lower mass range is primarily due to the fragmentation of larger masses during
the ionization process, as discussed extensively in Section 3.3.

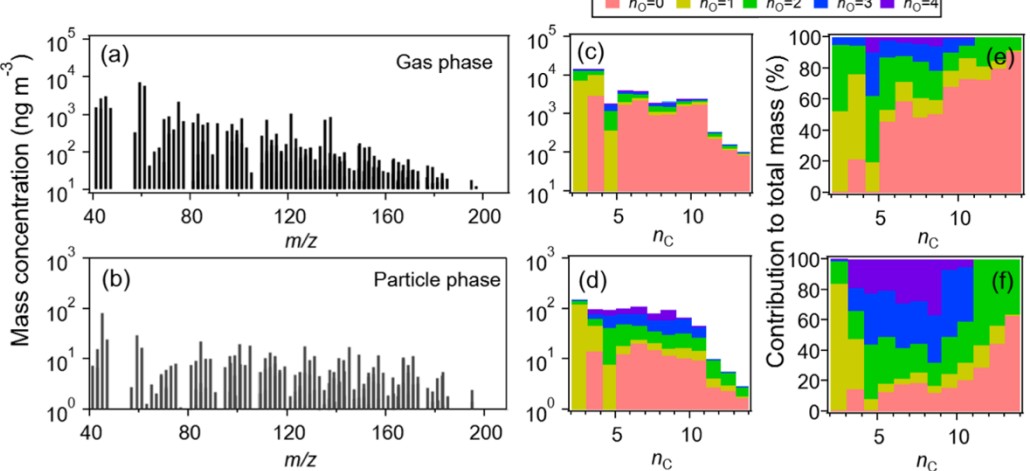


Figure 3. Background subtracted monthly-average PTR-ToF-MS mass spectra in the (a) gas
phase and (b) particle phase. Mass distributions of all identified species resolved by the carbon
and oxygen numbers ($n_C$ and $n_O$) in the (c) gas phase and (d) particle phase, as well as their
relative contribution to the total organic mass in the (e) gas phase and (f) particle phase.

Fig. 3 (a-b) shows the PTR-ToF-MS spectra of dominant ions averaged over the entire

campaign in both gas and particle phases. The mass concentrations of individual ions are in the
range of 7.9 – 7179.3 ng m$^{-3}$ in the gas phase and 0.6 – 82.7 ng m$^{-3}$ in the particle phase. A total
of 152 species (with > 60% data points above the PTR-ToF-MS detection limits) are identified,
contributing to ~ 69% and ~ 44%, respectively, of the total organic mass measured in the gas
and particle phase. The molecular distribution characterized by the carbon and oxygen of these
species is given in Fig. 3 (c-d). The most abundant species are characterized by a generic
formula of $C_xH_yO$ and $C_xH_yO_2$, resolving ~ 64% and ~ 46% in total of all identified species in
the gas and particle phase, respectively. Another dominant component in the gas phase is
hydrocarbon-like compounds ($C_xH_y$) (~ 27%), which contribute ~ 12% of the organic mass in
the particle phase. Species with higher oxygen numbers (> 2) contribute to a large fraction (~
42%) of the total particulate mass. These $C_xH_yO_{1-4}$ groups exhibit different diurnal cycles, as
shown in Fig. 4, reflecting their unique formation chemistry. The $C_xH_y$ group peaks in the early
morning rush hour and likely originates from primary traffic emissions. On the contrary, both
$C_xH_yO_3$ and $C_xH_yO_4$ groups peak at noon, suggesting a strong secondary formation source. The
diurnal trends for $C_xH_yO$ and $C_xH_yO_2$ groups are relatively flat during the day, likely indicative
of an intertwined primary emission and secondary formation processes.

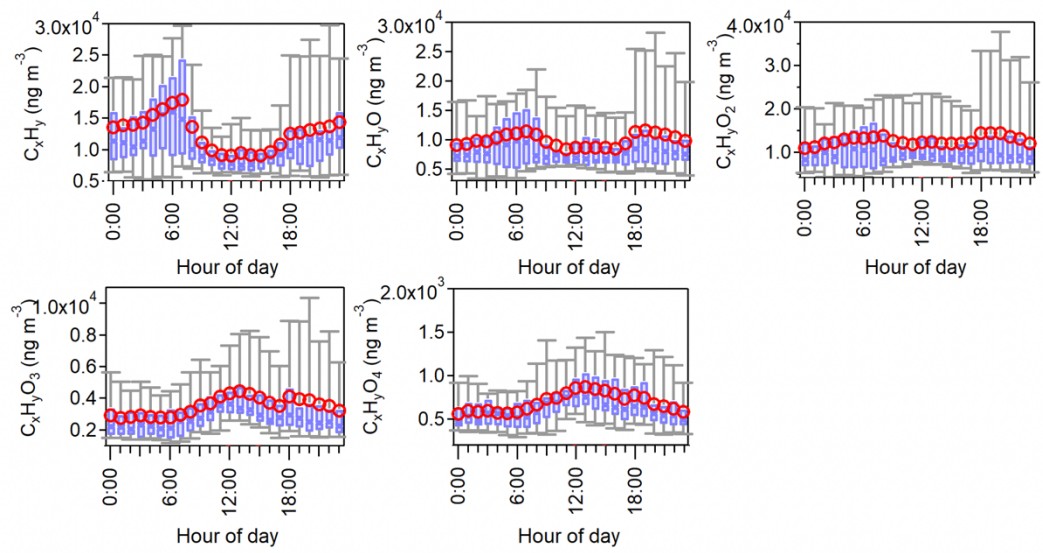


Figure 4. Diurnal variations of observed gas-phase species with a generic formula of $C_{2-13}H_{2-}$
$_{22}O_{0-4}$. Hourly average values (ng m$^{-3}$), together with 10$^{th}$, 25$^{th}$, 75$^{th}$, and 90$^{th}$ percentiles, are
also plotted.

**3.3. Measured vs. modeled gas-particle partitioning**
Fig. 5 shows the calculated particle-phase fraction ($F_p$) of the identified 152 species using
the CHARON-PTR-ToF-MS measurements in both phases. It is important to note that an
authentic standard is not required for the calculation of $F_p$ for any given species, because the
PTR sensitivity term is essentially canceled in the divisor function in Equations (6-8). Also
given in Fig. 5 is the simulated $F_p$ of the derived molecular formulas of all species identified
using the equilibrium partitioning theory, see method described in Section 2.4. Interestingly,
for oxidized species such as $C_xH_yO_4$, their measured $F_p$ agree reasonably with the simulations,
as shown in Fig. 5a. The $C_xH_yO_4$ group resides in the relatively higher mass range, and species
identified in this group are more likely actual compounds rather than fragments from larger
parent molecules. Even these species dessociate into lower mass ions during PTR ionization,
their calculated particle-phase fractions are unaffected by such fragmentation processes because
signals of parent ions decrease by the same extent upon fragmentation in both gas and particle
phases. As the oxygen number decreases, the measured $F_p$ values tend to deviate from the
simulations by up to several orders of magnitude. Note that these less oxidized compounds (e.g.,
$C_xH_y$) are mostly small molecules and they are highly unlikely present in the condensed phase
as closed-shell monomers (Pankow and Asher, 2008; Holzinger et al., 2010). Instead, they are
more likely fragments produced from the decomposition of larger molecules, which no
surprisingly favor partitioning in the particle phase. It is worth noting that the uptake of small
oxidized compounds on the aerosol aqueous phase does not significantly affect the overall
particle phase fraction of these compounds, see detailed calculations in Text S3. The
discrepancy in the measurement-model comparison underscores the importance of
understanding the fragmentation mechanism during PTR ionization when extracting molecular
features from the raw mass spectra.

Parent ion fragmentation has been widely observed in PTR-MS instruments (Pagonis et al.,

2019). Oxygenates exhibit trends in neutral losses of water or saturated alcohols. Here, we
apply a correction to the molecular formula of the 152 identified species by assuming that these
species are fragments produced from their parent precursors through the neutral losses of a
carboxyl group ($-CO_2$), a carbonyl group (-CO), a hydroxyl group ($-H_2O$), or an alcohol group
($-C_2H_6O$). By applying this correction, the modeled $F_p$ of a given species $C_xH_yO_z$ would actually
represent the particle-phase fraction of the its parent species $C_xH_yO_z \cdot CO_2$, $C_xH_yO_z \cdot CO$,
$C_xH_yO_z \cdot H_2O$, or $C_xH_yO_z \cdot C_2H_6O$. As shown in Fig. 5 (b-e), such a correction could significantly
increase the modeled $F_p$ values by several orders of magnitude. The assumption of neutral
losses of $CO_2$ or $C_2H_6O$ allows for much improved agreement between modeled vs. measured
$F_p$ values for less oxidized species. This implies that these small and less oxidized species are
likely fragments resulting from the decomposition of larger parent precursors. As our particle
enrichment test (details given in Section 3.1) has confirmed that the thermal desorption
temperature employed for particle evaporation does not lead to any intensive fragmentation,
therefore the collision-induced dissociation during the proton transfer reaction process becomes
the predominant process that produces fragments (Lindinger et al., 1998; Gueneron et al., 2015;
Gkatzelis et al., 2018b). Although the electric field applied to the drift tube is considered low
to moderate compared with most previous PTR-MS measurements ($E/N \sim 100$ Td in this study
vs. $E/N \sim 120 - 140$ Td commonly found in PTR-MS measurements) (Pagonis et al., 2019),
parent ion fragmentation was still widely observed here and complicated the mass spectra
interpretation and molecular feature extraction. While some recent CHARON measurements
employed lower electric field in the drift tube ($E/N \sim 60$ Td) (Leglise et al., 2019; Gkatzelis et
al., 2018a; Piel et al., 2019), such conditions could promote the formation of water cluster ions,
which increase with humidity and reduce the PTR sensitivity, and therefore are not ideally
suitable for our field measurements. A compromise solution would be using a moderate electric
field in the drift tube (e.g., $\sim 100$ Td) and meanwhile applying appropriate molecular corrections
to all ions detected in the mass spectra by considering possible neutral losses of small moieties.
Since in this study only the information of molecular formula is derived from the PTR-MS
spectra, we thus provide the lower and upper bound of the gas-particle partitioning corrections
owing to neutral losses of $H_2O$ and $CO_2$, respectively. In general, lower masses with higher
volatilities are subject to notable changes in the particle-phase fraction as a result of neutral
losses during the PTR ionization process, see detailed discussions in Text S2.

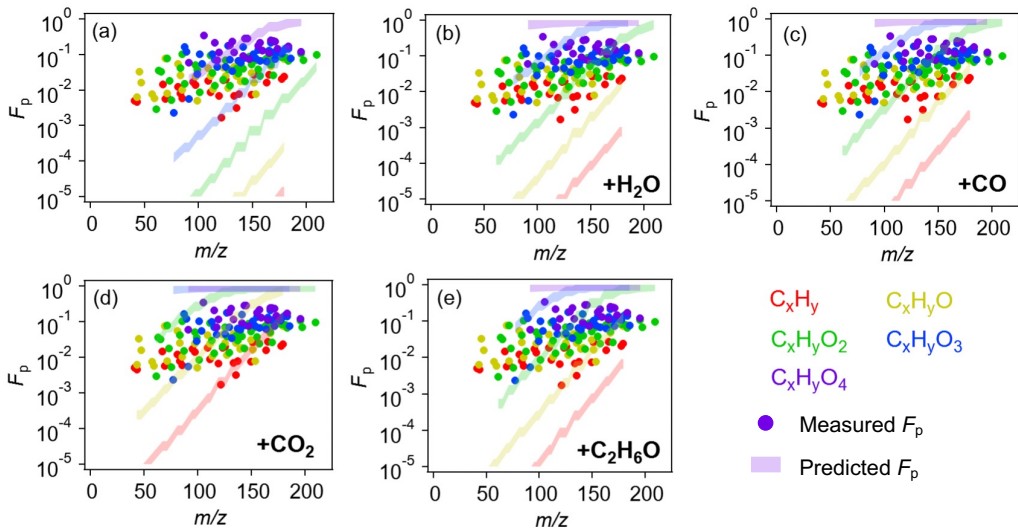


Figure 5. (a) Campaign average fraction of organic species in the particle phase ($F_p$) grouped
by the oxygen number. Solid markers represent the calculated $F_p$ based on the CHARON-PTR-
ToF-MS measurements. Colored shades represent the predicted $F_p$ of corresponding molecular
formulas. (b-e) Measured vs. predicted $F_p$ assuming the identified species are fragments of
corresponding parent compounds through neutral losses of $H_2O$, $CO$, $CO_2$, and $C_2H_6O$,
respectively (values are given in Table S5).

**4. Conclusions**
Recent studies have suggested that some of the model-measurement discrepancies in the
representation of ambient organic aerosol budget might be due to the nonequilibrium
gas/particle partitioning caused by kinetic limitations in the presence of glassy or semi-solid
phase (Perraud et al., 2012; Mai et al., 2015; Shiraiwa et al., 2013). It is therefore necessary to
validate whether the equilibrium partitioning theory could adequately describe the
condensation of semivolatile organic vapors onto atmospheric aerosols under ambient
conditions, and the accurate measurement of these SVOCs in both gas and particle phases is
the crucial prerequisite. In this study, we have employed the PTR-ToF-MS instrument coupled
to a CHARON inlet, together with a suite of complementary measurements, to characterize the
atmospheric partitioning behaviors of an array of SVOCs in an urban environment of East
China. Prior to the application to the field measurements, we first performed a series of
laboratory experiments to test whether the CHARON inlet is capable of sampling organic
molecules (including alcohols, carbonyls, and carboxylic acids) in their native states. With the
low pressure condition used in the CHARON inlet, a thermal desorption temperature less than
140 °C could adequately evaporate organic compounds with vapor pressure higher than $10^{-14}$
Pa while minimizing the thermal decomposition of labile functionalities. The auto-switch
function between the gas and particle mode with one single PTR-ToF-MS instrument could
monitor gaseous and particulate organic compounds in real time, thereby providing important
information of their partitioning behaviors in the ambient atmosphere. Particle-phase fractions
of a total of 152 organic species were derived from the CHARON-PTR-ToF-MS measurements
and further compared with model predictions using the instantaneous equilibrium partitioning
theory. While the model captured the particle-phase fraction of oxidized compounds (e.g.,
$C_xH_yO_{3-4}$), predictions of less oxidized compounds, notably the $C_xH_y$ family, differ from the
corresponding measurements by several orders of magnitude. Such a large discrepancy is very
likely caused by the intensive fragmentation of the parent organic compounds during the PTR
ionization process. Accounting for common fragmentation patterns in the simulations of gas-
particle partitioning, for example, neutral losses of -$CO_2$, -$CO$, -$H_2O$, or -$C_2H_6O$, could largely
improve the model-measurement agreement. Such corrections are very necessary towards an
accurate measurement of both particle- and gas-phase SVOCs using the CHARON-PTR-ToF-
MS instrument. Our study suggests the crucial importance of optimizing operation conditions
and understanding the fragmentation mechanism in the particle collection, vaporization, and
ionization processes in understanding the gas-particle partitioning of organic compounds using
any thermal desorption based aerosol measurement method.

*Data availability.* The data shown in the paper are available upon request from the
corresponding author.
*Author Contributions.* YP carried out experiments and measurements and drafted the
manuscript. HW and XZ designed the experimental studies, supervised the laboratory work and
wrote the manuscript. YG and SJ supported the ambient measurements. SZ, DH, PH, and SL
supported the data analysis. TC and CH supervised the scientific work. All authors have given
approval to the final version of the manuscript.
*Competing interests.* The authors declare no competing financial interest.
*Acknowledgements.* This research has been supported by the National Natural Science
Foundation of China (No. 42175135, No. 42175179), the National Key R&D Programm of
China (No. 2022YFE0136200) and the Shanghai Science and Technology Commission of the
Shanghai Municipality (No. 20ZR1447800).

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
