# Peer review of "Realtime measurement of phase partitioning of organic"

_Atmospheric Measurement Techniques, 2022_

## Author Comment (AC1)

**Response to comments from Anonymous Referee #1 (amt-2022-147):**

We thank the reviewer for the constructive and insightful comments, which significantly improved the quality of this work. Our point-by-point responses can be found below, with reviewer's comments in **black**, our responses in **blue**, alongside the relevant revisions to the manuscript in **red**.

This paper describes the use of an online CHARON inlet coupled to PTR-MS to measure the gas to particle fractions of a range of semi-volatile organic compounds. It combines experiments using a broad range of standard compounds and then expands this to look at real world data collected in Beijing. The main finding is that fragmentation of large molecules during the ionization process occurs and that this can have important implications when estimating gas-particle partitioning. This is a well-written paper and is well within scope AMT. The authors have provided the community with useful knowledge and some suggestions for how these difficulties could be overcome. I recommend publication after the following comments have been addressed.

1.    Figures: My main issue with this paper is a lack of clarity in the figures. Figure 1b is impossible to read with so many overlapping points. I think Figure 2 is meant to show the fragmentation pattern and how they change (or not in this case) at different temperatures. But the use of a log scale and a bar chart makes this very difficult to see. Also, each mass spectrum should have a labeled x-axis as these are all different.

A:    We have revised Figure 1b using a linear scale. Exact values and associated uncertainties were also provided in Table S3. We have also completely remade Figure 2, see below.

[Figure]

Figure 1. (a) Measured unitless enrichment factor (*EF*) of ammonium nitrate particles as a function of particle size in the 100 – 450 nm range. Grey markers represent all replicating measurements. The error bar denotes one standard deviation (1σ) of the average. (b) Ratios of PTR-ToF-MS signals (including both parent and fragment ions) to CPC counts (±1σ) at 300 nm for all organic standards studied. (c) *EF* (±1σ) of selected organic standards based on the calculated sensitivity.

[Figure]

Figure 2. Ratios of CHARON-PTR-ToF-MS signals (ncps) to CPC measurements (ppb) of all detected ions (including both parents and fragments) for a given pure organic standard tested under different desorption temperatures (70 – 140 °C), normalized to the corresponding ratios obtained at 140 °C. Ions with relative intensities less than 10% are excluded. Red markers represent the parent peaks. Colored shades represent the relative standard deviations at different temperatures (exact values are given in Table S4).

2.    I also think the use of a log scale in Figure 3c) and d) makes this very hard to interpret. It is difficult to see the relative percentages of the different groups using a log scale, where the pink colour is large but actually the % is relatively small. I would convert this to a linear scale or provide a scaled 100 % plot in addition to the one here.

A:    Revised accordingly. A scaled histogram has been added as subpanels (e) and (f).

[Figure]

Figure 3. Background subtracted monthly-averaged PTR-ToF-MS mass spectra in the (a) gas phase and (b) particle phase. Mass distributions of all identified species resolved by the carbon and oxygen numbers ($n_C$ and $n_O$) in the (c) gas phase and (d) particle phase, as well as their relative contribution to the total organic mass in the (e) gas phase and (f) particle phase.

Minor comments

3. Line 241: Why have the nitrophenols been excluded from the analysis?

A: A small number of nitrogen-containing organic species, potentially nitroaromatics, including $C_6H_5NO_3$ ($m/z$ 140), $C_7H_7NO_3$ ($m/z$ 154), $C_8H_9NO_3$ ($m/z$ 168), $C_6H_5NO_4$ ($m/z$ 156), and $C_7H_7NO_4$ ($m/z$ 170) have been resolved from the PTR mass spectra. Their total signals (counts per second, cps) only contribute to 0.2% of the total ion counts. Their total mass concentrations are also low, accounting for on average ~0.2% and ~0.7% of total measured mass in the gas and particle phase, respectively. Furthermore, the peaks of these nitrogen containing species are interfered by the $^{13}$C isotopes of the adjacent lower masses (see Fig. R1), introducing another layer of uncertainties to the quantification of these species.

[Figure]

Figure R1. High-resolution peak fitting for some potential nitrogen containing organic species from 5-hr average mass spectra measured.

4. Figure 1b: There are no blue or purple points at 70 C. Is there a reason for this? The figure legend needs more details. Is this normalized to the CPC counts at 300 nm diameter? Does the "PTR-TOF-MS" signal include the fragment ions?

A: Yes, this is normalized to the CPC counts at 300 nm diameter. The PTR-MS signals include all fragment ions. We have revised this figure accordingly.

5.    Line 330: Sucrose has a lower EF and this is attributed to dehydration. However, the light blue point in figure 1C (2,7-dihydroxynaphtalene) also looks to have a lower value but this compound doesn't fragment.

A:    The calculated enrichment factors ($EF$) of these two species under different temperatures are given in Table R1. The $EF$ values of sucrose have much larger variations. Furthermore, the $EF$ values of sucrose have much higher standard deviations at all temperatures due to fragmentation, see Table S3 given below. We have added corresponding discussions in the revised main text.

Table R1. The calculated enrichment factors ($EF$) of Sucrose and 2,7-dihydroxynaphtalene at different temperatures.

| Temperature (°C) | $EF$-Sucrose | $EF$-2,7-Dihydroxynaphthalene |
|---|---|---|
| 140 | 12 | 10 |
| 130 | 12 | 11 |
| 120 | 10 | 10 |
| 110 | 9 | 11 |
| 100 | 8 | 10 |
| 90 | 10 | 10 |
| 80 | 12 | 11 |
| 70 | 9 | 11 |

Table S3. Measured unitless enrichment factor ($EF$) and corresponding relative standard deviation (RSD) of the selected organic standards under different temperatures, Overall RSD $=\sqrt{RSD_{tem}^{2}+RSD_{each}^{2}}$.

| Standards | Molecular Formula | Mean signal in the 70-140 °C range (ncps ppb$^{-1}$) | $RSD_{tem}$ in the 70-140 °C range | $EF$ | $RSD_{each}$ at each temperature | Overall RSD |
|---|---|---|---|---|---|---|
| 2,7-Dihydroxynaphthalene | $C_{10}H_8O_2$ | 5551.44 | 3% | 10 | 11% | 12% |
| Levoglucosan | $C_6H_{10}O_5$ | 6687.43 | 3% | 14 | 13% | 13% |
| Phthalic acid | $C_8H_6O_4$ | 10611.46 | 5% | 13 | 11% | 12% |
| Vanillic acid | $C_8H_8O_4$ | 9694.58 | 1% | 16 | 7% | 7% |
| Cis-pinonic acid | $C_{10}H_{16}O_3$ | 5381.82 | 6% | 16 | 15% | 16% |
| 3-Methylbutane-1,2,3-tricarboxylic acid | $C_8H_{12}O_6$ | 6916.67 | 6% | 13 | 21% | 22% |
| 2-Pentadecanone | $C_{15}H_{30}O$ | 9370.66 | 3% | 13 | 19% | 19% |
| 1-Pentadecanol | $C_{15}H_{32}O$ | 12810.38 | 4% | 20 | 10% | 10% |
| Sucrose | $C_{12}H_{22}O_{11}$ | 6924.86 | 14% | 10 | 20% | 25% |

6.    Figure 2: I would have liked to have seen more discussion of the fragmentation data. What are the red labels on some peaks? They are not the molecular ion so I'm not sure why they are a different color.

A: We have remade Figure 2, also given below.

[Figure]

Figure 2. Ratios of CHARON-PTR-ToF-MS signals (ncps) to CPC measurements (ppb) of all detected ions (including both parents and fragments) for a given pure organic standard tested under different desorption temperatures (70 – 140 °C), normalized to the corresponding ratios obtained at 140 °C. Ions with relative intensities less than 10% are excluded. Red markers represent the parent peaks. Colored shades represent the relative standard deviations at different temperatures (exact values are given in Table S4).

As suggested, we have also added discussions of the fragmentation data in the revised manuscript, also given below:

"It has been recognized that species with one functional group follow certain fragmentation patterns during the PTR ionization process (Pagonis et al., 2019; Francis et al., 2007; Spanel et al., 1997; Tani et al., 2003; Spanel and Smith, 1997), such as dehydration of acids and alcohols. The observed dissociation of carboxylic acid standards used in this study, e.g., phthalic acid and 3-methylbutane-1,2,3-tricarboxylic acid, can be explained by this common fragmentation pattern. The fragmentation mechanism of muti-functionalized species is rather complicated and a number of fragments can be produced upon PTR ionization. Nevertheless, the identity and abundance of fragments from a given muti-functionalized species have been found comparable under the same PTR operation protocols (Leglise et al., 2019; Gkatzelis et al., 2018). For example, *cis*-pinonic acid yields the following fragments (main ions only and relative abundance in parentheses): *m/z* 71.049 (~28%), 115.075 (~36%), 167.108 (~19%), and 185.117 (~11%), which is comparable with an earlier study (Leglise et al., 2019): *m/z* 71.049 (~27%), 115.075 (~33%), 167.108 (~26%), and 185.117 (~14%) at 100 Td settings. "

7. Line 348: I think you need to make clear at the start of this section that you have moved to discussing the field data.

A: Revised as suggested:

"A month-long field dataset of particle- and gas-phase organic species was collected at hourly resolution using the CHARON-PTR-ToF-MS instrument."

8. Line 350-352: What does "agree very well mean"? In figure S5 you have R values. Can these be added to figure S4?

A: We have added the correlation coefficients for these two measurements. Strong correlation was found for aromatics and oxygenated compounds, with correlation coefficients higher than ~0.8. Acetone is an exception: the GC measurements, after a period of instrument down from 13/11 5:00 to 14/11 3:00, are substantially higher than the PTR measurements, although the observed trends by both instruments are still identical. We have revised the manuscript accordingly, also given below.

"For the gas phase, quantitative measurements of a suite of VOCs by GC-MS/FID, including benzene, toluene, styrene, $C_8$ and $C_9$ aromatics, acrolein, and $C_4$, $C_5$, and $C_6$ ketones, agree well with corresponding PTR-ToF-MS measurements, as shown in Fig. S6."

[Figure]

Figure S6. Comparison of month-long mass concentrations ($\mu g\ m^{-3}$) of gaseous organic compounds measured by PTR-ToF-MS (PTR) and GC-MS/FID (GC).

9. Line 355: Figure S5 shows a comparison of specific compounds not total carbon mass.

A: Revised. It should be the overall molecular mass.

10. Line 356: This comparison is interesting – it seems that when the mass is low towards the end of the measurement period, there is a larger discrepancy between the two methods. Do you have any suggestions for why this is?

A: We think the discrepancy between the two measurements at low aerosol mass loadings might be attributed to different protocols used for background subtraction. For PTR, backgrounds were measured periodically and then all collected data were background subtracted. For TAG, blank

samples were collected every two days to ensure the aerosol sampling and desorption system was not contaminated by any residuals. The actual measurement data were derived from the calibration curves of selected authentic standard without any background subtraction. (Wang et al., 2020; Zhu et al., 2021).

11. Line 358: I cannot see a plot of CHARON derived total OA or its comparison to AMS data?

A: A comparison between AMS measured OA and CHARON derived total OA mass has been added in Fig. R2 (Fig. S9), also given below. Despite a reasonable correlation in time series, CHARON reported much less OA mass than AMS, for three reasons:

First, the CHARON derived OA mass only includes compounds with $m/z$ < 300. Our observation that the majority of detected organic particulate compounds are distributed in the lower m/z range is consistent with a number of earlier studies (Salvador et al., 2016; Holzinger et al., 2010b; Holzinger et al., 2013).

Second, a sensitivity of ~933 ncps/ppb, close to that of acetone (978 ncps/ppb) was used to derive the concentrations of all detected masses. This sensitivity value was calculated using a reaction rate constant of $3 \times 10^{-9}\,cm^3\,s^{-1}\,molecule^{-1}$, which has been suggested as the optimal estimate of the average rate constant for aerosol species (Holzinger et al., 2013; Holzinger et al., 2010a), and was considered as an upper limit of the PTR sensitivity towards various organic molecules. Therefore, the derived concentration of each organic species in the particle phase represents the lower limit of their actual particulate concentration in the ambient air.

Third, in order to calculate the gas-particle partitioning based on the CHARON-PTR measurements, the concentration of a given species needs to be above the detection limit in both gas and particle phases. This data selection criterion inevitably excludes certain organic masses that are only present in a single phase. For the identified 152 species, their PTR sensitivities were experimentally determined using calibration gases or calculated using the rate coefficients estimated from the polarizability and dipole moment of the molecule (Sekimoto et al., 2017). These species in total account for 60% of the total PTR signals, as shown by the red circles in Fig. R2.

[Figure]

Figure R2. Scatter plot of organic aerosol masses measured by CHARON-PTR-ToF-MS vs. AMS. Black markers represent the total mass concentration of all nominal masses below 300 Th and red markers represent the total mass concentrations of the identified 152 species in the particle phase.

12. Line 394: Is there any explanation for why the O4 group is closest or is it just by chance? In

Figure 4, adding additional functional groups to the O4 species makes the agreement worse. Perhaps these are less prone to fragmentation?

A:   The $C_xH_yO_4$ group resides in the relatively higher mass range, and species identified in this group are more likely actual compounds rather than fragments from larger parent molecules. Even these species dissociate into lower mass ions during PTR ionization, their calculated particle-phase fractions are unaffected by such fragmentation processes because signals of parent ions decrease by the same extent upon fragmentation in both gas and particle phases. We have added corresponding discussions in the revised main text.

13. Figure 4: I think you should present the data in Figure 4 in a table in addition to the plots. This would be more useful for readers.

A: As suggested, we have presented the data used in Figure 4 in Table S5 in detail.

Table S5. Campaign average fraction of the measured ($F_{p,m}$) and predicted ($F_{p,p}$) organic species in the particle phase grouped by the oxygen number. The predicted $F_p$ were also corrected assuming the identified species are fragments of corresponding parent compounds by neutral losses of $H_2O$, CO, $CO_2$, and $C_2H_6O$, respectively.

| Detected ion | Mass | Class | $F_{p,m}$ | $F_{p,p}$ | $F_{p,p}$ (+H$_2$O) | $F_{p,p}$ (+CO) | $F_{p,p}$ (+CO$_2$) | $F_{p,p}$ (+C$_2$H$_6$O) |
|---|---|---|---|---|---|---|---|---|
| $C_3H_4H^+$ | 41.039 | $C_xH_y$ | $5.08\times10^{-3}$ | $2.59\times10^{-10}$ | $1.83\times10^{-8}$ | $5.11\times10^{-8}$ | $4.88\times10^{-6}$ | $1.46\times10^{-7}$ |
| $C_2H_2OH^+$ | 43.018 | $C_xH_yO$ | $6.30\times10^{-3}$ | $6.88\times10^{-9}$ | $8.68\times10^{-7}$ | $1.96\times10^{-6}$ | $2.59\times10^{-4}$ | $4.88\times10^{-6}$ |
| $C_3H_6H^+$ | 43.054 | $C_xH_y$ | $4.64\times10^{-3}$ | $2.59\times10^{-10}$ | $1.83\times10^{-8}$ | $5.11\times10^{-8}$ | $4.88\times10^{-6}$ | $1.46\times10^{-7}$ |
| $C_2H_4OH^+$ | 45.033 | $C_xH_yO$ | $3.34\times10^{-2}$ | $6.88\times10^{-9}$ | $8.68\times10^{-7}$ | $1.96\times10^{-6}$ | $2.59\times10^{-4}$ | $4.88\times10^{-6}$ |
| $C_2H_6OH^+$ | 47.049 | $C_xH_yO$ | $1.62\times10^{-2}$ | $6.88\times10^{-9}$ | $8.68\times10^{-7}$ | $1.96\times10^{-6}$ | $2.59\times10^{-4}$ | $4.88\times10^{-6}$ |
| $C_3H_4OH^+$ | 57.033 | $C_xH_yO$ | $7.75\times10^{-3}$ | $1.83\times10^{-8}$ | $1.96\times10^{-6}$ | $4.88\times10^{-6}$ | $5.75\times10^{-4}$ | $1.28\times10^{-5}$ |
| $C_3H_6OH^+$ | 59.049 | $C_xH_yO$ | $5.87\times10^{-3}$ | $1.83\times10^{-8}$ | $1.96\times10^{-6}$ | $4.88\times10^{-6}$ | $5.75\times10^{-4}$ | $1.28\times10^{-5}$ |
| $C_2H_4O_2H^+$ | 61.028 | $C_xH_yO_2$ | $2.97\times10^{-3}$ | $8.66\times10^{-7}$ | $1.31\times10^{-4}$ | $2.59\times10^{-4}$ | $3.70\times10^{-2}$ | $5.75\times10^{-4}$ |
| $C_2H_6O_2H^+$ | 63.044 | $C_xH_yO_2$ | $3.62\times10^{-2}$ | $8.66\times10^{-7}$ | $1.31\times10^{-4}$ | $2.59\times10^{-4}$ | $3.70\times10^{-2}$ | $5.75\times10^{-4}$ |
| $C_2H_8O_2H^+$ | 65.060 | $C_xH_yO_2$ | $3.26\times10^{-2}$ | $8.66\times10^{-7}$ | $1.31\times10^{-4}$ | $2.59\times10^{-4}$ | $3.70\times10^{-2}$ | $5.75\times10^{-4}$ |
| $C_5H_6H^+$ | 67.054 | $C_xH_y$ | $1.46\times10^{-2}$ | $2.31\times10^{-9}$ | $1.46\times10^{-7}$ | $4.21\times10^{-7}$ | $3.45\times10^{-5}$ | $1.23\times10^{-6}$ |
| $C_4H_4OH^+$ | 69.033 | $C_xH_yO$ | $7.75\times10^{-2}$ | $5.10\times10^{-8}$ | $4.88\times10^{-6}$ | $1.28\times10^{-5}$ | $1.37\times10^{-3}$ | $3.45\times10^{-5}$ |
| $C_5H_8H^+$ | 69.070 | $C_xH_y$ | $5.73\times10^{-3}$ | $2.31\times10^{-9}$ | $1.46\times10^{-7}$ | $4.21\times10^{-7}$ | $3.45\times10^{-5}$ | $1.23\times10^{-6}$ |
| $C_3H_2O_2H^+$ | 71.013 | $C_xH_yO_2$ | $7.81\times10^{-2}$ | $1.96\times10^{-6}$ | $2.59\times10^{-4}$ | $5.75\times10^{-4}$ | $7.18\times10^{-2}$ | $1.37\times10^{-3}$ |
| $C_4H_6OH^+$ | 71.049 | $C_xH_yO$ | $8.01\times10^{-3}$ | $5.10\times10^{-8}$ | $4.88\times10^{-6}$ | $1.28\times10^{-5}$ | $1.37\times10^{-3}$ | $3.45\times10^{-5}$ |
| $C_5H_{10}H^+$ | 71.086 | $C_xH_y$ | $6.52\times10^{-3}$ | $2.31\times10^{-9}$ | $1.46\times10^{-7}$ | $4.21\times10^{-7}$ | $3.45\times10^{-5}$ | $1.23\times10^{-6}$ |
| $C_3H_4O_2H^+$ | 73.028 | $C_xH_yO_2$ | $1.82\times10^{-2}$ | $1.96\times10^{-6}$ | $2.59\times10^{-4}$ | $5.75\times10^{-4}$ | $7.18\times10^{-2}$ | $1.37\times10^{-3}$ |
| $C_3H_6O_2H^+$ | 75.044 | $C_xH_yO_2$ | $5.28\times10^{-3}$ | $1.96\times10^{-6}$ | $2.59\times10^{-4}$ | $5.75\times10^{-4}$ | $7.18\times10^{-2}$ | $1.37\times10^{-3}$ |
| $C_2H_4O_3H^+$ | 77.023 | $C_xH_yO_3$ | $2.33\times10^{-3}$ | $1.31\times10^{-4}$ | $2.13\times10^{-2}$ | $3.70\times10^{-2}$ | $8.60\times10^{-1}$ | $7.18\times10^{-2}$ |
| $C_5H_4OH^+$ | 81.033 | $C_xH_yO$ | $5.53\times10^{-2}$ | $1.45\times10^{-7}$ | $1.28\times10^{-5}$ | $3.45\times10^{-5}$ | $3.44\times10^{-3}$ | $9.55\times10^{-5}$ |
| $C_6H_8H^+$ | 81.070 | $C_xH_y$ | $1.24\times10^{-2}$ | $6.88\times10^{-9}$ | $4.21\times10^{-7}$ | $1.23\times10^{-6}$ | $9.55\times10^{-5}$ | $3.59\times10^{-6}$ |
| $C_4H_2O_2H^+$ | 83.013 | $C_xH_yO_2$ | $4.21\times10^{-2}$ | $4.87\times10^{-6}$ | $5.75\times10^{-4}$ | $1.37\times10^{-3}$ | $1.45\times10^{-1}$ | $3.44\times10^{-3}$ |
| $C_5H_6OH^+$ | 83.049 | $C_xH_yO$ | $2.59\times10^{-2}$ | $1.45\times10^{-7}$ | $1.28\times10^{-5}$ | $3.45\times10^{-5}$ | $3.44\times10^{-3}$ | $9.55\times10^{-5}$ |
| $C_6H_{10}H^+$ | 83.086 | $C_xH_y$ | $7.98\times10^{-3}$ | $6.88\times10^{-9}$ | $4.21\times10^{-7}$ | $1.23\times10^{-6}$ | $9.55\times10^{-5}$ | $3.59\times10^{-6}$ |
| $C_4H_4O_2H^+$ | 85.028 | $C_xH_yO_2$ | $1.03\times10^{-1}$ | $4.87\times10^{-6}$ | $5.75\times10^{-4}$ | $1.37\times10^{-3}$ | $1.45\times10^{-1}$ | $3.44\times10^{-3}$ |

| Ion | Mass | Group | | | | | | |
|---|---|---|---|---|---|---|---|---|
| $C_5H_8OH^+$ | 85.065 | $C_xH_yO$ | $1.02\times10^{-2}$ | $1.45\times10^{-7}$ | $1.28\times10^{-5}$ | $3.45\times10^{-5}$ | $3.44\times10^{-3}$ | $9.55\times10^{-5}$ |
| $C_6H_{12}H^+$ | 85.101 | $C_xH_y$ | $5.96\times10^{-3}$ | $6.88\times10^{-9}$ | $4.21\times10^{-7}$ | $1.23\times10^{-6}$ | $9.55\times10^{-5}$ | $3.59\times10^{-6}$ |
| $C_3H_2O_3H^+$ | 87.008 | $C_xH_yO_3$ | $1.66\times10^{-1}$ | $2.59\times10^{-4}$ | $3.70\times10^{-2}$ | $7.18\times10^{-2}$ | $9.19\times10^{-1}$ | $1.45\times10^{-1}$ |
| $C_4H_6O_2H^+$ | 87.044 | $C_xH_yO_2$ | $1.51\times10^{-2}$ | $4.87\times10^{-6}$ | $5.75\times10^{-4}$ | $1.37\times10^{-3}$ | $1.45\times10^{-1}$ | $3.44\times10^{-3}$ |
| $C_3H_4O_3H^+$ | 89.023 | $C_xH_yO_3$ | $1.20\times10^{-1}$ | $2.59\times10^{-4}$ | $3.70\times10^{-2}$ | $7.18\times10^{-2}$ | $9.19\times10^{-1}$ | $1.45\times10^{-1}$ |
| $C_2H_2O_4H^+$ | 91.003 | $C_xH_yO_4$ | $5.99\times10^{-2}$ | $2.10\times10^{-2}$ | $7.92\times10^{-1}$ | $8.60\times10^{-1}$ | $9.99\times10^{-1}$ | $9.19\times10^{-1}$ |
| $C_3H_6O_3H^+$ | 91.039 | $C_xH_yO_3$ | $5.58\times10^{-3}$ | $2.59\times10^{-4}$ | $3.70\times10^{-2}$ | $7.18\times10^{-2}$ | $9.19\times10^{-1}$ | $1.45\times10^{-1}$ |
| $C_7H_{10}H^+$ | 95.086 | $C_xH_y$ | $1.71\times10^{-2}$ | $2.05\times10^{-8}$ | $1.23\times10^{-6}$ | $3.59\times10^{-6}$ | $2.68\times10^{-4}$ | $1.06\times10^{-5}$ |
| $C_5H_4O_2H^+$ | 97.028 | $C_xH_yO_2$ | $5.11\times10^{-2}$ | $1.28\times10^{-5}$ | $1.37\times10^{-3}$ | $3.44\times10^{-3}$ | $2.84\times10^{-1}$ | $8.90\times10^{-3}$ |
| $C_6H_8OH^+$ | 97.065 | $C_xH_yO$ | $2.47\times10^{-2}$ | $4.20\times10^{-7}$ | $3.45\times10^{-5}$ | $9.55\times10^{-5}$ | $8.90\times10^{-3}$ | $2.68\times10^{-4}$ |
| $C_7H_{12}H^+$ | 97.101 | $C_xH_y$ | $1.40\times10^{-2}$ | $2.05\times10^{-8}$ | $1.23\times10^{-6}$ | $3.59\times10^{-6}$ | $2.68\times10^{-4}$ | $1.06\times10^{-5}$ |
| $C_4H_2O_3H^+$ | 99.008 | $C_xH_yO_3$ | $2.85\times10^{-2}$ | $5.74\times10^{-4}$ | $7.18\times10^{-2}$ | $1.45\times10^{-1}$ | $9.58\times10^{-1}$ | $2.84\times10^{-1}$ |
| $C_5H_6O_2H^+$ | 99.044 | $C_xH_yO_2$ | $3.67\times10^{-2}$ | $1.28\times10^{-5}$ | $1.37\times10^{-3}$ | $3.44\times10^{-3}$ | $2.84\times10^{-1}$ | $8.90\times10^{-3}$ |
| $C_6H_{10}OH^+$ | 99.080 | $C_xH_yO$ | $5.13\times10^{-3}$ | $4.20\times10^{-7}$ | $3.45\times10^{-5}$ | $9.55\times10^{-5}$ | $8.90\times10^{-3}$ | $2.68\times10^{-4}$ |
| $C_4H_4O_3H^+$ | 101.023 | $C_xH_yO_3$ | $1.02\times10^{-1}$ | $5.74\times10^{-4}$ | $7.18\times10^{-2}$ | $1.45\times10^{-1}$ | $9.58\times10^{-1}$ | $2.84\times10^{-1}$ |
| $C_5H_8O_2H^+$ | 101.060 | $C_xH_yO_2$ | $8.67\times10^{-3}$ | $1.28\times10^{-5}$ | $1.37\times10^{-3}$ | $3.44\times10^{-3}$ | $2.84\times10^{-1}$ | $8.90\times10^{-3}$ |
| $C_4H_6O_3H^+$ | 103.039 | $C_xH_yO_3$ | $5.49\times10^{-2}$ | $5.74\times10^{-4}$ | $7.18\times10^{-2}$ | $1.45\times10^{-1}$ | $9.58\times10^{-1}$ | $2.84\times10^{-1}$ |
| $C_3H_4O_4H^+$ | 105.018 | $C_xH_yO_4$ | $3.48\times10^{-1}$ | $3.62\times10^{-2}$ | $8.60\times10^{-1}$ | $9.19\times10^{-1}$ | $9.99\times10^{-1}$ | $9.58\times10^{-1}$ |
| $C_7H_8OH^+$ | 109.065 | $C_xH_yO$ | $4.26\times10^{-2}$ | $1.22\times10^{-6}$ | $9.55\times10^{-5}$ | $2.68\times10^{-4}$ | $2.34\times10^{-2}$ | $7.61\times10^{-4}$ |
| $C_8H_{12}H^+$ | 109.101 | $C_xH_y$ | $1.84\times10^{-2}$ | $6.13\times10^{-8}$ | $3.59\times10^{-6}$ | $1.06\times10^{-5}$ | $7.61\times10^{-4}$ | $3.11\times10^{-5}$ |
| $C_6H_6O_2H^+$ | 111.044 | $C_xH_yO_2$ | $9.37\times10^{-2}$ | $3.45\times10^{-5}$ | $3.44\times10^{-3}$ | $8.90\times10^{-3}$ | $4.92\times10^{-1}$ | $2.34\times10^{-2}$ |
| $C_7H_{10}OH^+$ | 111.080 | $C_xH_yO$ | $2.49\times10^{-2}$ | $1.22\times10^{-6}$ | $9.55\times10^{-5}$ | $2.68\times10^{-4}$ | $2.34\times10^{-2}$ | $7.61\times10^{-4}$ |
| $C_8H_{14}H^+$ | 111.117 | $C_xH_y$ | $8.78\times10^{-3}$ | $6.13\times10^{-8}$ | $3.59\times10^{-6}$ | $1.06\times10^{-5}$ | $7.61\times10^{-4}$ | $3.11\times10^{-5}$ |
| $C_5H_4O_3H^+$ | 113.023 | $C_xH_yO_3$ | $1.01\times10^{-1}$ | $1.37\times10^{-3}$ | $1.45\times10^{-1}$ | $2.84\times10^{-1}$ | $9.81\times10^{-1}$ | $4.92\times10^{-1}$ |
| $C_6H_8O_2H^+$ | 113.060 | $C_xH_yO_2$ | $3.02\times10^{-2}$ | $3.45\times10^{-5}$ | $3.44\times10^{-3}$ | $8.90\times10^{-3}$ | $4.92\times10^{-1}$ | $2.34\times10^{-2}$ |
| $C_7H_{12}OH^+$ | 113.096 | $C_xH_yO$ | $1.07\times10^{-2}$ | $1.22\times10^{-6}$ | $9.55\times10^{-5}$ | $2.68\times10^{-4}$ | $2.34\times10^{-2}$ | $7.61\times10^{-4}$ |
| $C_4H_2O_4H^+$ | 115.003 | $C_xH_yO_4$ | $2.13\times10^{-1}$ | $6.90\times10^{-2}$ | $9.19\times10^{-1}$ | $9.58\times10^{-1}$ | $1.00$ | $9.81\times10^{-1}$ |
| $C_5H_6O_3H^+$ | 115.039 | $C_xH_yO_3$ | $7.16\times10^{-2}$ | $1.37\times10^{-3}$ | $1.45\times10^{-1}$ | $2.84\times10^{-1}$ | $9.81\times10^{-1}$ | $4.92\times10^{-1}$ |
| $C_6H_{10}O_2H^+$ | 115.075 | $C_xH_yO_2$ | $1.34\times10^{-2}$ | $3.45\times10^{-5}$ | $3.44\times10^{-3}$ | $8.90\times10^{-3}$ | $4.92\times10^{-1}$ | $2.34\times10^{-2}$ |
| $C_4H_4O_4H^+$ | 117.018 | $C_xH_yO_4$ | $6.98\times10^{-2}$ | $6.90\times10^{-2}$ | $9.19\times10^{-1}$ | $9.58\times10^{-1}$ | $1.00$ | $9.81\times10^{-1}$ |
| $C_5H_8O_3H^+$ | 117.055 | $C_xH_yO_3$ | $4.48\times10^{-2}$ | $1.37\times10^{-3}$ | $1.45\times10^{-1}$ | $2.84\times10^{-1}$ | $9.81\times10^{-1}$ | $4.92\times10^{-1}$ |
| $C_4H_6O_4H^+$ | 119.034 | $C_xH_yO_4$ | $1.17\times10^{-1}$ | $6.90\times10^{-2}$ | $9.19\times10^{-1}$ | $9.58\times10^{-1}$ | $1.00$ | $9.81\times10^{-1}$ |
| $C_9H_{10}H^+$ | 119.086 | $C_xH_y$ | $6.79\times10^{-3}$ | $1.83\times10^{-7}$ | $1.06\times10^{-5}$ | $3.11\times10^{-5}$ | $2.17\times10^{-3}$ | $9.19\times10^{-5}$ |
| $C_9H_{12}H^+$ | 121.101 | $C_xH_y$ | $1.72\times10^{-3}$ | $1.83\times10^{-7}$ | $1.06\times10^{-5}$ | $3.11\times10^{-5}$ | $2.17\times10^{-3}$ | $9.19\times10^{-5}$ |
| $C_7H_6O_2H^+$ | 123.044 | $C_xH_yO_2$ | $2.03\times10^{-2}$ | $9.53\times10^{-5}$ | $8.90\times10^{-3}$ | $2.34\times10^{-2}$ | $7.10\times10^{-1}$ | $6.11\times10^{-2}$ |
| $C_8H_{10}OH^+$ | 123.080 | $C_xH_yO$ | $4.38\times10^{-2}$ | $3.58\times10^{-6}$ | $2.68\times10^{-4}$ | $7.61\times10^{-4}$ | $6.11\times10^{-2}$ | $2.17\times10^{-3}$ |
| $C_9H_{14}H^+$ | 123.117 | $C_xH_y$ | $1.88\times10^{-2}$ | $1.83\times10^{-7}$ | $1.06\times10^{-5}$ | $3.11\times10^{-5}$ | $2.17\times10^{-3}$ | $9.19\times10^{-5}$ |
| $C_6H_4O_3H^+$ | 125.023 | $C_xH_yO_3$ | $4.61\times10^{-2}$ | $3.43\times10^{-3}$ | $2.84\times10^{-1}$ | $4.92\times10^{-1}$ | $9.91\times10^{-1}$ | $7.10\times10^{-1}$ |
| $C_7H_8O_2H^+$ | 125.060 | $C_xH_yO_2$ | $6.95\times10^{-2}$ | $9.53\times10^{-5}$ | $8.90\times10^{-3}$ | $2.34\times10^{-2}$ | $7.10\times10^{-1}$ | $6.11\times10^{-2}$ |
| $C_8H_{12}OH^+$ | 125.096 | $C_xH_yO$ | $2.55\times10^{-2}$ | $3.58\times10^{-6}$ | $2.68\times10^{-4}$ | $7.61\times10^{-4}$ | $6.11\times10^{-2}$ | $2.17\times10^{-3}$ |
| $C_9H_{16}H^+$ | 125.132 | $C_xH_y$ | $1.99\times10^{-2}$ | $1.83\times10^{-7}$ | $1.06\times10^{-5}$ | $3.11\times10^{-5}$ | $2.17\times10^{-3}$ | $9.19\times10^{-5}$ |
| $C_6H_6O_3H^+$ | 127.039 | $C_xH_yO_3$ | $1.81\times10^{-1}$ | $3.43\times10^{-3}$ | $2.84\times10^{-1}$ | $4.92\times10^{-1}$ | $9.91\times10^{-1}$ | $7.10\times10^{-1}$ |
| $C_7H_{10}O_2H^+$ | 127.075 | $C_xH_yO_2$ | $3.87\times10^{-2}$ | $9.53\times10^{-5}$ | $8.90\times10^{-3}$ | $2.34\times10^{-2}$ | $7.10\times10^{-1}$ | $6.11\times10^{-2}$ |
| $C_8H_{14}OH^+$ | 127.112 | $C_xH_yO$ | $1.17\times10^{-2}$ | $3.58\times10^{-6}$ | $2.68\times10^{-4}$ | $7.61\times10^{-4}$ | $6.11\times10^{-2}$ | $2.17\times10^{-3}$ |

| Formula | Mass | Type | | | | | |
|---|---|---|---|---|---|---|---|
| $C_5H_4O_4H^+$ | 129.018 | $C_xH_yO_4$ | $1.86\times10^{-1}$ | $1.36\times10^{-1}$ | $9.58\times10^{-1}$ | $9.81\times10^{-1}$ | $1.00$ | $9.91\times10^{-1}$ |
| $C_6H_8O_3H^+$ | 129.055 | $C_xH_yO_3$ | $7.18\times10^{-2}$ | $3.43\times10^{-3}$ | $2.84\times10^{-1}$ | $4.92\times10^{-1}$ | $9.91\times10^{-1}$ | $7.10\times10^{-1}$ |
| $C_7H_{12}O_2H^+$ | 129.091 | $C_xH_yO_2$ | $1.36\times10^{-2}$ | $9.53\times10^{-5}$ | $8.90\times10^{-3}$ | $2.34\times10^{-2}$ | $7.10\times10^{-1}$ | $6.11\times10^{-2}$ |
| $C_5H_6O_4H^+$ | 131.034 | $C_xH_yO_4$ | $2.38\times10^{-1}$ | $1.36\times10^{-1}$ | $9.58\times10^{-1}$ | $9.81\times10^{-1}$ | $1.00$ | $9.91\times10^{-1}$ |
| $C_6H_{10}O_3H^+$ | 131.070 | $C_xH_yO_3$ | $3.49\times10^{-2}$ | $3.43\times10^{-3}$ | $2.84\times10^{-1}$ | $4.92\times10^{-1}$ | $9.91\times10^{-1}$ | $7.10\times10^{-1}$ |
| $C_5H_8O_4H^+$ | 133.050 | $C_xH_yO_4$ | $1.17\times10^{-1}$ | $1.36\times10^{-1}$ | $9.58\times10^{-1}$ | $9.81\times10^{-1}$ | $1.00$ | $9.91\times10^{-1}$ |
| $C_9H_8OH^+$ | 133.065 | $C_xH_yO$ | $4.52\times10^{-2}$ | $1.05\times10^{-5}$ | $7.61\times10^{-4}$ | $2.17\times10^{-3}$ | $1.52\times10^{-1}$ | $6.24\times10^{-3}$ |
| $C_{10}H_{12}H^+$ | 133.101 | $C_xH_y$ | $7.02\times10^{-3}$ | $5.47\times10^{-7}$ | $3.11\times10^{-5}$ | $9.19\times10^{-5}$ | $6.24\times10^{-3}$ | $2.72\times10^{-4}$ |
| $C_8H_6O_2H^+$ | 135.044 | $C_xH_yO_2$ | $4.00\times10^{-2}$ | $2.68\times10^{-4}$ | $2.34\times10^{-2}$ | $6.11\times10^{-2}$ | $8.64\times10^{-1}$ | $1.52\times10^{-1}$ |
| $C_9H_{10}OH^+$ | 135.080 | $C_xH_yO$ | $2.53\times10^{-2}$ | $1.05\times10^{-5}$ | $7.61\times10^{-4}$ | $2.17\times10^{-3}$ | $1.52\times10^{-1}$ | $6.24\times10^{-3}$ |
| $C_{10}H_{14}H^+$ | 135.117 | $C_xH_y$ | $3.18\times10^{-3}$ | $5.47\times10^{-7}$ | $3.11\times10^{-5}$ | $9.19\times10^{-5}$ | $6.24\times10^{-3}$ | $2.72\times10^{-4}$ |
| $C_7H_4O_3H^+$ | 137.023 | $C_xH_yO_3$ | $6.61\times10^{-2}$ | $8.84\times10^{-3}$ | $4.92\times10^{-1}$ | $7.10\times10^{-1}$ | $9.96\times10^{-1}$ | $8.64\times10^{-1}$ |
| $C_8H_8O_2H^+$ | 137.060 | $C_xH_yO_2$ | $3.00\times10^{-2}$ | $2.68\times10^{-4}$ | $2.34\times10^{-2}$ | $6.11\times10^{-2}$ | $8.64\times10^{-1}$ | $1.52\times10^{-1}$ |
| $C_9H_{12}OH^+$ | 137.096 | $C_xH_yO$ | $3.99\times10^{-2}$ | $1.05\times10^{-5}$ | $7.61\times10^{-4}$ | $2.17\times10^{-3}$ | $1.52\times10^{-1}$ | $6.24\times10^{-3}$ |
| $C_{10}H_{16}H^+$ | 137.132 | $C_xH_y$ | $7.83\times10^{-3}$ | $5.47\times10^{-7}$ | $3.11\times10^{-5}$ | $9.19\times10^{-5}$ | $6.24\times10^{-3}$ | $2.72\times10^{-4}$ |
| $C_7H_6O_3H^+$ | 139.039 | $C_xH_yO_3$ | $5.92\times10^{-2}$ | $8.84\times10^{-3}$ | $4.92\times10^{-1}$ | $7.10\times10^{-1}$ | $9.96\times10^{-1}$ | $8.64\times10^{-1}$ |
| $C_8H_{10}O_2H^+$ | 139.075 | $C_xH_yO_2$ | $5.51\times10^{-2}$ | $2.68\times10^{-4}$ | $2.34\times10^{-2}$ | $6.11\times10^{-2}$ | $8.64\times10^{-1}$ | $1.52\times10^{-1}$ |
| $C_9H_{14}OH^+$ | 139.112 | $C_xH_yO$ | $1.74\times10^{-2}$ | $1.05\times10^{-5}$ | $7.61\times10^{-4}$ | $2.17\times10^{-3}$ | $1.52\times10^{-1}$ | $6.24\times10^{-3}$ |
| $C_6H_4O_4H^+$ | 141.018 | $C_xH_yO_4$ | $9.21\times10^{-2}$ | $2.57\times10^{-1}$ | $9.81\times10^{-1}$ | $9.91\times10^{-1}$ | $1.00$ | $9.96\times10^{-1}$ |
| $C_7H_8O_3H^+$ | 141.055 | $C_xH_yO_3$ | $1.62\times10^{-1}$ | $8.84\times10^{-3}$ | $4.92\times10^{-1}$ | $7.10\times10^{-1}$ | $9.96\times10^{-1}$ | $8.64\times10^{-1}$ |
| $C_8H_{12}O_2H^+$ | 141.091 | $C_xH_yO_2$ | $3.99\times10^{-2}$ | $2.68\times10^{-4}$ | $2.34\times10^{-2}$ | $6.11\times10^{-2}$ | $8.64\times10^{-1}$ | $1.52\times10^{-1}$ |
| $C_9H_{16}OH^+$ | 141.127 | $C_xH_yO$ | $1.84\times10^{-2}$ | $1.05\times10^{-5}$ | $7.61\times10^{-4}$ | $2.17\times10^{-3}$ | $1.52\times10^{-1}$ | $6.24\times10^{-3}$ |
| $C_6H_6O_4H^+$ | 143.034 | $C_xH_yO_4$ | $2.81\times10^{-1}$ | $2.57\times10^{-1}$ | $9.81\times10^{-1}$ | $9.91\times10^{-1}$ | $1.00$ | $9.96\times10^{-1}$ |
| $C_7H_{10}O_3H^+$ | 143.070 | $C_xH_yO_3$ | $8.03\times10^{-2}$ | $8.84\times10^{-3}$ | $4.92\times10^{-1}$ | $7.10\times10^{-1}$ | $9.96\times10^{-1}$ | $8.64\times10^{-1}$ |
| $C_8H_{14}O_2H^+$ | 143.107 | $C_xH_yO_2$ | $1.76\times10^{-2}$ | $2.68\times10^{-4}$ | $2.34\times10^{-2}$ | $6.11\times10^{-2}$ | $8.64\times10^{-1}$ | $1.52\times10^{-1}$ |
| $C_6H_8O_4H^+$ | 145.050 | $C_xH_yO_4$ | $2.86\times10^{-1}$ | $2.57\times10^{-1}$ | $9.81\times10^{-1}$ | $9.91\times10^{-1}$ | $1.00$ | $9.96\times10^{-1}$ |
| $C_9H_6O_2H^+$ | 147.044 | $C_xH_yO_2$ | $8.89\times10^{-2}$ | $7.59\times10^{-4}$ | $6.11\times10^{-2}$ | $1.52\times10^{-1}$ | $9.44\times10^{-1}$ | $3.33\times10^{-1}$ |
| $C_6H_{10}O_4H^+$ | 147.065 | $C_xH_yO_4$ | $7.55\times10^{-2}$ | $2.57\times10^{-1}$ | $9.81\times10^{-1}$ | $9.91\times10^{-1}$ | $1.00$ | $9.96\times10^{-1}$ |
| $C_8H_4O_3H^+$ | 149.023 | $C_xH_yO_3$ | $8.18\times10^{-2}$ | $2.30\times10^{-2}$ | $7.10\times10^{-1}$ | $8.64\times10^{-1}$ | $9.99\times10^{-1}$ | $9.44\times10^{-1}$ |
| $C_9H_8O_2H^+$ | 149.060 | $C_xH_yO_2$ | $5.26\times10^{-2}$ | $7.59\times10^{-4}$ | $6.11\times10^{-2}$ | $1.52\times10^{-1}$ | $9.44\times10^{-1}$ | $3.33\times10^{-1}$ |
| $C_{10}H_{12}OH^+$ | 149.096 | $C_xH_yO$ | $3.10\times10^{-2}$ | $3.11\times10^{-5}$ | $2.17\times10^{-3}$ | $6.24\times10^{-3}$ | $3.33\times10^{-1}$ | $1.79\times10^{-2}$ |
| $C_{11}H_{16}H^+$ | 149.132 | $C_xH_y$ | $7.56\times10^{-3}$ | $1.63\times10^{-6}$ | $9.19\times10^{-5}$ | $2.72\times10^{-4}$ | $1.79\times10^{-2}$ | $8.05\times10^{-4}$ |
| $C_8H_6O_3H^+$ | 151.039 | $C_xH_yO_3$ | $1.03\times10^{-1}$ | $2.30\times10^{-2}$ | $7.10\times10^{-1}$ | $8.64\times10^{-1}$ | $9.99\times10^{-1}$ | $9.44\times10^{-1}$ |
| $C_9H_{10}O_2H^+$ | 151.075 | $C_xH_yO_2$ | $8.29\times10^{-2}$ | $7.59\times10^{-4}$ | $6.11\times10^{-2}$ | $1.52\times10^{-1}$ | $9.44\times10^{-1}$ | $3.33\times10^{-1}$ |
| $C_{10}H_{14}OH^+$ | 151.112 | $C_xH_yO$ | $1.65\times10^{-2}$ | $3.11\times10^{-5}$ | $2.17\times10^{-3}$ | $6.24\times10^{-3}$ | $3.33\times10^{-1}$ | $1.79\times10^{-2}$ |
| $C_{11}H_{18}H^+$ | 151.148 | $C_xH_y$ | $2.68\times10^{-2}$ | $1.63\times10^{-6}$ | $9.19\times10^{-5}$ | $2.72\times10^{-4}$ | $1.79\times10^{-2}$ | $8.05\times10^{-4}$ |
| $C_8H_8O_3H^+$ | 153.055 | $C_xH_yO_3$ | $7.72\times10^{-2}$ | $2.30\times10^{-2}$ | $7.10\times10^{-1}$ | $8.64\times10^{-1}$ | $9.99\times10^{-1}$ | $9.44\times10^{-1}$ |
| $C_9H_{12}O_2H^+$ | 153.091 | $C_xH_yO_2$ | $6.97\times10^{-2}$ | $7.59\times10^{-4}$ | $6.11\times10^{-2}$ | $1.52\times10^{-1}$ | $9.44\times10^{-1}$ | $3.33\times10^{-1}$ |
| $C_{10}H_{16}OH^+$ | 153.127 | $C_xH_yO$ | $7.90\times10^{-3}$ | $3.11\times10^{-5}$ | $2.17\times10^{-3}$ | $6.24\times10^{-3}$ | $3.33\times10^{-1}$ | $1.79\times10^{-2}$ |
| $C_7H_6O_4H^+$ | 155.034 | $C_xH_yO_4$ | $1.79\times10^{-1}$ | $4.38\times10^{-1}$ | $9.91\times10^{-1}$ | $9.96\times10^{-1}$ | $1.00$ | $9.99\times10^{-1}$ |
| $C_8H_{10}O_3H^+$ | 155.070 | $C_xH_yO_3$ | $1.36\times10^{-1}$ | $2.30\times10^{-2}$ | $7.10\times10^{-1}$ | $8.64\times10^{-1}$ | $9.99\times10^{-1}$ | $9.44\times10^{-1}$ |
| $C_9H_{14}O_2H^+$ | 155.107 | $C_xH_yO_2$ | $3.82\times10^{-2}$ | $7.59\times10^{-4}$ | $6.11\times10^{-2}$ | $1.52\times10^{-1}$ | $9.44\times10^{-1}$ | $3.33\times10^{-1}$ |
| $C_7H_8O_4H^+$ | 157.050 | $C_xH_yO_4$ | $2.20\times10^{-1}$ | $4.38\times10^{-1}$ | $9.91\times10^{-1}$ | $9.96\times10^{-1}$ | $1.00$ | $9.99\times10^{-1}$ |
| $C_8H_{12}O_3H^+$ | 157.086 | $C_xH_yO_3$ | $6.79\times10^{-2}$ | $2.30\times10^{-2}$ | $7.10\times10^{-1}$ | $8.64\times10^{-1}$ | $9.99\times10^{-1}$ | $9.44\times10^{-1}$ |

| | | | | | | | |
|---|---|---|---|---|---|---|---|
| $C_9H_{16}O_2H^+$ | 157.122 | $C_xH_yO_2$ | $2.49\times10^{-2}$ | $7.59\times10^{-4}$ | $6.11\times10^{-2}$ | $1.52\times10^{-1}$ | $9.44\times10^{-1}$ | $3.33\times10^{-1}$ |
| $C_7H_{10}O_4H^+$ | 159.065 | $C_xH_yO_4$ | $1.31\times10^{-1}$ | $4.38\times10^{-1}$ | $9.91\times10^{-1}$ | $9.96\times10^{-1}$ | 1.00 | $9.99\times10^{-1}$ |
| $C_{10}H_8O_2H^+$ | 161.060 | $C_xH_yO_2$ | $7.07\times10^{-2}$ | $2.17\times10^{-3}$ | $1.52\times10^{-1}$ | $3.33\times10^{-1}$ | $9.78\times10^{-1}$ | $5.85\times10^{-1}$ |
| $C_9H_6O_3H^+$ | 163.039 | $C_xH_yO_3$ | $6.36\times10^{-2}$ | $5.90\times10^{-2}$ | $8.64\times10^{-1}$ | $9.44\times10^{-1}$ | $9.99\times10^{-1}$ | $9.78\times10^{-1}$ |
| $C_{10}H_{10}O_2H^+$ | 163.075 | $C_xH_yO_2$ | $7.99\times10^{-2}$ | $2.17\times10^{-3}$ | $1.52\times10^{-1}$ | $3.33\times10^{-1}$ | $9.78\times10^{-1}$ | $5.85\times10^{-1}$ |
| $C_{12}H_{18}H^+$ | 163.148 | $C_xH_y$ | $1.61\times10^{-2}$ | $4.87\times10^{-6}$ | $2.72\times10^{-4}$ | $8.05\times10^{-4}$ | $5.02\times10^{-2}$ | $2.38\times10^{-3}$ |
| $C_8H_4O_4H^+$ | 165.018 | $C_xH_yO_4$ | $1.12\times10^{-1}$ | $6.43\times10^{-1}$ | $9.96\times10^{-1}$ | $9.99\times10^{-1}$ | 1.00 | $9.99\times10^{-1}$ |
| $C_9H_8O_3H^+$ | 165.055 | $C_xH_yO_3$ | $1.31\times10^{-1}$ | $5.90\times10^{-2}$ | $8.64\times10^{-1}$ | $9.44\times10^{-1}$ | $9.99\times10^{-1}$ | $9.78\times10^{-1}$ |
| $C_{10}H_{12}O_2H^+$ | 165.091 | $C_xH_yO_2$ | $7.85\times10^{-2}$ | $2.17\times10^{-3}$ | $1.52\times10^{-1}$ | $3.33\times10^{-1}$ | $9.78\times10^{-1}$ | $5.85\times10^{-1}$ |
| $C_{11}H_{16}OH^+$ | 165.127 | $C_xH_yO$ | $5.81\times10^{-2}$ | $9.18\times10^{-5}$ | $6.24\times10^{-3}$ | $1.79\times10^{-2}$ | $5.85\times10^{-1}$ | $5.02\times10^{-2}$ |
| $C_{12}H_{20}H^+$ | 165.164 | $C_xH_y$ | $2.77\times10^{-2}$ | $4.87\times10^{-6}$ | $2.72\times10^{-4}$ | $8.05\times10^{-4}$ | $5.02\times10^{-2}$ | $2.38\times10^{-3}$ |
| $C_8H_6O_4H^+$ | 167.034 | $C_xH_yO_4$ | $2.50\times10^{-1}$ | $6.43\times10^{-1}$ | $9.96\times10^{-1}$ | $9.99\times10^{-1}$ | 1.00 | $9.99\times10^{-1}$ |
| $C_9H_{10}O_3H^+$ | 167.070 | $C_xH_yO_3$ | $2.14\times10^{-1}$ | $5.90\times10^{-2}$ | $8.64\times10^{-1}$ | $9.44\times10^{-1}$ | $9.99\times10^{-1}$ | $9.78\times10^{-1}$ |
| $C_{10}H_{14}O_2H^+$ | 167.107 | $C_xH_yO_2$ | $5.21\times10^{-2}$ | $2.17\times10^{-3}$ | $1.52\times10^{-1}$ | $3.33\times10^{-1}$ | $9.78\times10^{-1}$ | $5.85\times10^{-1}$ |
| $C_{11}H_{18}OH^+$ | 167.143 | $C_xH_yO$ | $4.79\times10^{-2}$ | $9.18\times10^{-5}$ | $6.24\times10^{-3}$ | $1.79\times10^{-2}$ | $5.85\times10^{-1}$ | $5.02\times10^{-2}$ |
| $C_8H_8O_4H^+$ | 169.050 | $C_xH_yO_4$ | $2.43\times10^{-1}$ | $6.43\times10^{-1}$ | $9.96\times10^{-1}$ | $9.99\times10^{-1}$ | 1.00 | $9.99\times10^{-1}$ |
| $C_9H_{12}O_3H^+$ | 169.086 | $C_xH_yO_3$ | $1.14\times10^{-1}$ | $5.90\times10^{-2}$ | $8.64\times10^{-1}$ | $9.44\times10^{-1}$ | $9.99\times10^{-1}$ | $9.78\times10^{-1}$ |
| $C_{10}H_{16}O_2H^+$ | 169.122 | $C_xH_yO_2$ | $2.64\times10^{-2}$ | $2.17\times10^{-3}$ | $1.52\times10^{-1}$ | $3.33\times10^{-1}$ | $9.78\times10^{-1}$ | $5.85\times10^{-1}$ |
| $C_8H_{10}O_4H^+$ | 171.065 | $C_xH_yO_4$ | $2.32\times10^{-1}$ | $6.43\times10^{-1}$ | $9.96\times10^{-1}$ | $9.99\times10^{-1}$ | 1.00 | $9.99\times10^{-1}$ |
| $C_9H_{14}O_3H^+$ | 171.102 | $C_xH_yO_3$ | $7.56\times10^{-2}$ | $5.90\times10^{-2}$ | $8.64\times10^{-1}$ | $9.44\times10^{-1}$ | $9.99\times10^{-1}$ | $9.78\times10^{-1}$ |
| $C_8H_{12}O_4H^+$ | 173.081 | $C_xH_yO_4$ | $1.49\times10^{-1}$ | $6.43\times10^{-1}$ | $9.96\times10^{-1}$ | $9.99\times10^{-1}$ | 1.00 | $9.99\times10^{-1}$ |
| $C_9H_{16}O_3H^+$ | 173.117 | $C_xH_yO_3$ | $1.16\times10^{-1}$ | $5.90\times10^{-2}$ | $8.64\times10^{-1}$ | $9.44\times10^{-1}$ | $9.99\times10^{-1}$ | $9.78\times10^{-1}$ |
| $C_{10}H_8O_3H^+$ | 177.055 | $C_xH_yO_3$ | $1.08\times10^{-1}$ | $1.42\times10^{-1}$ | $9.44\times10^{-1}$ | $9.78\times10^{-1}$ | 1.00 | $9.92\times10^{-1}$ |
| $C_{11}H_{12}O_2H^+$ | 177.091 | $C_xH_yO_2$ | $8.45\times10^{-2}$ | $6.21\times10^{-3}$ | $3.33\times10^{-1}$ | $5.85\times10^{-1}$ | $9.92\times10^{-1}$ | $8.00\times10^{-1}$ |
| $C_{13}H_{20}H^+$ | 177.164 | $C_xH_y$ | $2.07\times10^{-2}$ | $1.45\times10^{-5}$ | $8.05\times10^{-4}$ | $2.38\times10^{-3}$ | $1.34\times10^{-1}$ | $7.04\times10^{-3}$ |
| $C_{10}H_{10}O_3H^+$ | 179.070 | $C_xH_yO_3$ | $1.35\times10^{-1}$ | $1.42\times10^{-1}$ | $9.44\times10^{-1}$ | $9.78\times10^{-1}$ | 1.00 | $9.92\times10^{-1}$ |
| $C_{11}H_{14}O_2H^+$ | 179.107 | $C_xH_yO_2$ | $8.47\times10^{-2}$ | $6.21\times10^{-3}$ | $3.33\times10^{-1}$ | $5.85\times10^{-1}$ | $9.92\times10^{-1}$ | $8.00\times10^{-1}$ |
| $C_{12}H_{18}OH^+$ | 179.143 | $C_xH_yO$ | $7.47\times10^{-2}$ | $2.71\times10^{-4}$ | $1.79\times10^{-2}$ | $5.02\times10^{-2}$ | $8.00\times10^{-1}$ | $1.34\times10^{-1}$ |
| $C_{13}H_{22}H^+$ | 179.179 | $C_xH_y$ | $2.40\times10^{-2}$ | $1.45\times10^{-5}$ | $8.05\times10^{-4}$ | $2.38\times10^{-3}$ | $1.34\times10^{-1}$ | $7.04\times10^{-3}$ |
| $C_{10}H_{12}O_3H^+$ | 181.086 | $C_xH_yO_3$ | $1.53\times10^{-1}$ | $1.42\times10^{-1}$ | $9.44\times10^{-1}$ | $9.78\times10^{-1}$ | 1.00 | $9.92\times10^{-1}$ |
| $C_{11}H_{16}O_2H^+$ | 181.122 | $C_xH_yO_2$ | $8.12\times10^{-2}$ | $6.21\times10^{-3}$ | $3.33\times10^{-1}$ | $5.85\times10^{-1}$ | $9.92\times10^{-1}$ | $8.00\times10^{-1}$ |
| $C_9H_{10}O_4H^+$ | 183.065 | $C_xH_yO_4$ | $1.73\times10^{-1}$ | $8.12\times10^{-1}$ | $9.99\times10^{-1}$ | $9.99\times10^{-1}$ | 1.00 | 1.00 |
| $C_{10}H_{14}O_3H^+$ | 183.102 | $C_xH_yO_3$ | $1.44\times10^{-1}$ | $1.42\times10^{-1}$ | $9.44\times10^{-1}$ | $9.78\times10^{-1}$ | 1.00 | $9.92\times10^{-1}$ |
| $C_{11}H_{18}O_2H^+$ | 183.138 | $C_xH_yO_2$ | $6.64\times10^{-2}$ | $6.21\times10^{-3}$ | $3.33\times10^{-1}$ | $5.85\times10^{-1}$ | $9.92\times10^{-1}$ | $8.00\times10^{-1}$ |
| $C_{10}H_{16}O_3H^+$ | 185.117 | $C_xH_yO_3$ | $7.71\times10^{-2}$ | $1.42\times10^{-1}$ | $9.44\times10^{-1}$ | $9.78\times10^{-1}$ | 1.00 | $9.92\times10^{-1}$ |
| $C_{10}H_{10}O_4H^+$ | 195.065 | $C_xH_yO_4$ | $1.21\times10^{-1}$ | $9.15\times10^{-1}$ | $9.99\times10^{-1}$ | 1.00 | 1.00 | 1.00 |
| $C_{12}H_{18}O_2H^+$ | 195.138 | $C_xH_yO_2$ | $1.09\times10^{-1}$ | $1.76\times10^{-2}$ | $5.85\times10^{-1}$ | $8.00\times10^{-1}$ | $9.97\times10^{-1}$ | $9.19\times10^{-1}$ |
| $C_{12}H_{20}O_2H^+$ | 197.154 | $C_xH_yO_2$ | $7.00\times10^{-2}$ | $1.76\times10^{-2}$ | $5.85\times10^{-1}$ | $8.00\times10^{-1}$ | $9.97\times10^{-1}$ | $9.19\times10^{-1}$ |
| $C_{13}H_{20}O_2H^+$ | 209.154 | $C_xH_yO_2$ | $9.76\times10^{-2}$ | $4.88\times10^{-2}$ | $8.00\times10^{-1}$ | $9.19\times10^{-1}$ | $9.99\times10^{-1}$ | $9.70\times10^{-1}$ |

14. Line 430: I agree with your final comment here but I think you need to be clear that currently this isn't possible as you can't predict what the parent molecule was.

A: Agreed. The fragmentation mechanism of a given parent molecule under a certain collision energy depends ultimately on the structure of that molecule.

As only the information of molecular formula is derived from the PTR-MS spectra, we thus can provide the lower and upper bound of the gas-particle partitioning corrections by considering common neutral loss patterns. The most common fragmentation mechanism during the PTR ionization process includes the neutral losses of a carboxyl group ($-CO_2$), a carbonyl group ($-CO$), a hydroxyl group ($-H_2O$), or an alcohol group ($-C_2H_6O$), with predicted volatilities ($C^*$) of $1.58\times10^7$, $2.51\times10^9$, $3.76\times10^9$ and $1.06\times10^9$ ($\mu g\ m^{-3}$), respectively. If a species $C_xH_yO_z$ loses a $H_2O$ group, for example, and becomes $C_xH_yO_z-H_2O$, then its saturation mass concentration ($C^*$) will increase by around 100 times accordingly. For example, for the species detected as $C_6H_6O_3$ in the PTR-MS mass spectra, it is unknown whether this species is a real compound present in the air or a fragment of its parent compound $C_6H_6O_3+H_2O$. If it is the latter, then the model predicted $C^*$ based on the derived formula of $C_6H_6O_3$ would be 99 times higher than its actual $C^*$, and as a result, the model predicted particle-phase fraction ($F_{p,mod}$ of $C_6H_6O_3$ = 0.003) would be significantly lower than its actual particle-phase fraction ($F_{p,mod}$ of $C_6H_8O_4$ = 0.284).

Our calculations show that neutral losses of $H_2O$ and $CO_2$ give the lower and upper limit, respectively, of the predicted fraction of all compounds investigated in this study. Specifically, for $C_{3-13}H_{4-22}$, $C_{2-12}H_{2-18}O$, $C_{2-11}H_{4-14}O_2$, $C_{2-10}H_{4-16}O_3$ and $C_{2-10}H_{2-10}O_4$, the predicted $F_p$ values increase by around $5.85\times10^1\sim1.27\times10^4$, $8.33\times10^1\sim1.97\times10^4$, $9.57\times10^1\sim1.33\times10^4$, $61.2\sim774$ and $6.03\sim6.87$, respectively. It is no surprise that lower masses with higher volatilities are subject to significant changes in the particle-phase fraction as a result of neutral losses during the PTR ionization process.

We have added corresponding discussions in the revised main text.

"Since in this study only the information of molecular formula is derived from the PTR-MS spectra, we thus provide the lower and upper bound of the gas-particle partitioning corrections owing to neutral losses of $H_2O$ and $CO_2$, respectively. In general, lower masses with higher volatilities are subject to notable changes in the particle-phase fraction as a result of neutral losses during the PTR ionization process, see detailed discussions in Text S1."

[Figure]

Figure R3. Predicted $F_p$ as a function of $C^*$ of (a-e) different groups on a log scale assuming the identified species are fragments of corresponding parent compounds through neutral losses of $H_2O$, $CO$, $CO_2$, and $C_2H_6O$.

**References**

[revised manuscript text omitted]

---

## Author Response (AR2)

We thank the reviewers for the constructive and insightful comments, which significantly improved the quality of this work. Our point-by-point responses can be found below, with reviewer comments in **black**, our responses in blue, alongside the relevant revisions to the manuscript in red.

Generally, I think the authors answered my comments and suggestions very thoroughly and well, leading to a great improvement on an already very good manuscript. However, some of the thorough responses didn't lead to any additions to the manuscript text. I think they should not stay buried in the referee response file, but added to the manuscript or SI, because other readers may have the same questions, and the authors put a lot of work into them:

1. The answer to question 2 should be added to the method section or the SI.

A: As suggested, we have added the answers to the section 2.2.3 and supplement file (Text S1).

Line 187 "Calibration standards with higher molecular weight were excluded because we only considered ion masses below 200 amu from the field measurement for the study of gas-particle partitioning, see discussions given in Text S1."

Line 203 "Uncertainties associated with the addition of the low-mass filter have been accounted for in the regression of individual transmission efficiency measurements on corresponding mass to charge ratios. The overall relative standard deviations were less than 15%."

2. The answer to question 6 should also be added to the manuscript, although it could of course be shortened.

A: As suggested, we have added it in the revised manuscript.

Line 148 "This temperature was chosen to ensure all the unknowns observed in the field can be evaporated effectively while maintaining relatively intact molecular structures, see more details in Section 3.1."

3. One sentence on the answer to question 17 should be added to the manuscript.

A: As suggested, we have added it in the revised manuscript.

Line 488 "It is worth noting that the uptake of small oxidized compounds on the aerosol aqueous phase does not significantly affect the overall particle phase fraction of these compounds, see detailed calculations in Text S3."

4. L. 127 in the manuscript: I think this sentence is missing info on the low mass cutoff that you discussed in the response and show in the Supplement. I.e., I suggest you add "…transmits ions more efficiently but leads to a low-mass cutoff (Fig. S4)."

A: Revised as suggested.

Line 127 "The PTR-ToF-MS instrument used here is equipped with a radio frequency (RF)-only quadrupole ion guide that transmits ions more efficiently (PTR-QiTOF, Ionicon Analytik Inc) but results in a low-mass cutoff (Fig. S4)."